# Contribution of expanded marine sulfur chemistry to the seasonal variability of DMS oxidation products and size-resolved sulfate aerosol

Linia Tashmim[1], William C. Porter[1], Qianjie Chen[2], Becky Alexander[3], Charles H. Fite[4], Christopher D. Holmes[4], Jeffrey R. Pierce[5], Betty Croft[6], and Sakiko Ishino[7]

[1]Department of Environmental Sciences, University of California, Riverside, CA, USA
[2]Department of Civil and Environmental Engineering, The Hong Kong Polytechnic University, Hong Kong, China
[3]Department of Atmospheric Sciences, University of Washington, Seattle, WA, USA
[4]Department of Earth, Ocean and Atmospheric Science, Florida State University, Tallahassee, FL, USA
[5]Department of Atmospheric Science, Colorado State University, Fort Collins, CO, USA
[6]Department of Physics and Atmospheric Science, Dalhousie University, Halifax, Nova Scotia, Canada
[7]Institute of Nature and Environmental Technology, Kanazawa University, Japan

Correspondence: Linia Tashmim (ltash001@ucr.edu) and William C. Porter (william.porter@ucr.edu)

**Abstract.** Marine emissions of dimethyl sulfide (DMS) and the subsequent formation of its oxidation products methane sulfonic acid (MSA) and sulfuric acid ($H_2SO_4$) are well-known natural precursors of atmospheric aerosols, contributing to particle mass and cloud formation over ocean and coastal regions. Despite a long-recognized and well-studied role in the marine troposphere, DMS oxidation chemistry remains a work in progress within many current air quality and climate models, with recent advances exploring heterogeneous chemistry and uncovering previously unknown intermediate species. With the identification of additional DMS oxidation pathways and intermediate species influencing its eventual fate, it is important to understand the impact of these pathways on the overall sulfate aerosol budget and aerosol size distribution. In this work, we update and evaluate the DMS oxidation mechanism of the chemical transport model GEOS-Chem by implementing expanded DMS oxidation pathways into the model. These updates include gas- and aqueous-phase reactions, the formation of the intermediates dimethyl sulfoxide (DMSO) and methane sulphinic acid (MSIA), as well as cloud loss and aerosol uptake of the recently quantified intermediate hydroperoxymethyl thioformate (HPMTF). We find that this updated mechanism collectively decreases the global mean surface-layer gas-phase sulfur dioxide ($SO_2$) mixing ratio by 40% and enhances sulfate aerosol ($SO_4^{2-}$) mixing ratio by 17%. We further perform sensitivity analyses exploring the contribution of cloud loss and aerosol uptake of HPMTF to the overall sulfur budget. Comparing modeled concentrations to available observations we find improved biases relative to previous studies. To quantify impacts of these chemistry updates on global particle size distributions and mass concentration we use the TOMAS aerosol microphysics module coupled to GEOS-Chem, finding changes in particle formation and growth affect the size distribution of aerosol. With this new DMS-oxidation scheme the global annual mean surface layer number concentration of particles with diameters smaller than 80 nm decreases by 16.8%, with cloud loss processes related to HPMTF mostly responsible for this reduction. However, global annual mean number of particles larger than 80 nm increases by 3.8% suggesting that the new scheme promotes seasonal particle growth to these sizes capable of acting as cloud condensation nuclei (CCN).

## 1 Introduction

Dimethyl sulfide (DMS: $CH_3SCH_3$) is the most abundant biological source of sulfate aerosol and has a significant influence on Earth's radiation budget and climate due to its contribution to atmospheric marine particle (Charlson et al., 1987; Fung et al., 2022). In the atmosphere, DMS reacts with hydroxyl radical (OH), nitrate radical ($NO_3$), ozone ($O_3$) and various halogen species (e.g., chlorine (Cl) and bromine oxide (BrO)), primarily forming sulfur dioxide ($SO_2$) and methyl sulfonic acid (MSA: $CH_3SO_3H$) (Chen et al., 2018; Faloona, 2009; Hoffmann et al., 2016). These oxidation products are considered key influences on the formation and evolution of natural aerosols and clouds along with their associated climate impacts, especially in the marine boundary layer (MBL) (Carslaw et al., 2013; Sipilä et al., 2010; Schobesberger et al., 2013; Thomas et al., 2010; von Glasow and Crutzen, 2004). $SO_2$ and MSA formed by DMS oxidation can be deposited on Earth surface or further oxidize affecting the size distribution of aerosol and cloud microphysics (Leaitch et al., 2013; Wollesen de Jonge et al., 2021). $SO_2$ can either oxidize in the gas-phase by reaction with the OH radical forming $H_2SO_4$, which can participate in nucleation and early growth of particles in the atmosphere, or it can be taken up by cloud droplets and undergo aqueous phase oxidation by reaction with $H_2O_2$, $O_3$ and $O_2$ catalyzed by transition metals (Mn, Fe) forming $SO_4^{2-}$ and generally only contributing to the growth of aerosol particles (Hoyle et al., 2016; Kulmala, 2003; Alexander et al., 2009). The hypohalous acids (HOBr, HOCl, HOI) also plays significant role in aqueous-phase sulfate production in the marine boundary layer (MBL) (Chen et al., 2016; Sherwen et al., 2016b). Recent studies have highlighted the importance of natural aerosols originating from DMS oxidation and their contribution to the uncertainty of aerosol radiative forcing in climate models (Carslaw et al., 2013; Fung et al., 2022; Rosati et al., 2022; Novak et al., 2021, 2022). Since DMS-derived aerosol is a major source of uncertainty in estimating the global natural aerosol burden and associated aerosol indirect radiative forcing, a more accurate representation of DMS oxidation and particle formation processes is an important step towards improved Earth system and climate modeling.

Although the chemistry of DMS oxidation has been previously studied in great detail, known uncertainties and omissions in the current mechanism remain in current air quality and chemical transport models (Barnes et al., 2006; Fung et al., 2022; Hoffmann et al., 2016, 2021). Furthermore, while increasingly complex and experimentally validated mechanisms are under ongoing development, DMS oxidation processes in many current chemical transport models continue to be represented through simplified gas-phase reactions with the tropospheric oxidants OH and $NO_3$, producing the two major oxidation products $SO_2$ and MSA at a fixed ratio as shown in R1-R3 in Table 1 (Chen et al., 2018; Chin et al., 1996; Veres et al., 2020). This type of simplified mechanism neglects the formation and loss of important intermediates such as dimethyl sulfoxide (DMSO: $CH_3SOCH_3$), methane sulphinic acid (MSIA: $CH_3SO_2H$) and the recently discovered oxidation product hydroperoxymethyl thioformate (HPMTF: $HOOCH_2SCHO$) (Berndt et al., 2019; Veres et al., 2020; Wu et al., 2015; Khan et al., 2021).

These omissions can have major consequences on product yields of DMS oxidation, thereby affecting the aerosol burdens. For example, the OH-addition pathway of DMS forms DMSO and MSIA as the intermediates, which has been identified as a dominant source of MSA via their aqueous-phase oxidation, and a fraction of that MSA subsequently undergoes aqueous-phase oxidation to form sulfate aerosol (Chen et al., 2018; Ishino et al., 2021; Zhu et al., 2006; von Glasow and Crutzen, 2004). Previous studies suggest that BrO contributes to 8 – 30% of total DMS

loss, highlighting the importance of this pathway as well (Breider et al., 2010; Boucher et al., 2003;
Chen et al., 2018; Khan et al., 2016). More recent experimental and laboratory studies have
confirmed the formation of methylthiomethyl peroxy radicals ($CH_3CH_2OO$; abbreviated as MSP
or MTMP) from the H-abstraction channel of OH oxidation, which can subsequently lead to a
series of rapid intramolecular H-shift isomerization reactions, ultimately resulting in the formation
of the stable intermediate HPMTF (Berndt et al., 2019; Veres et al., 2020; Vermeuel et al., 2020;
Wu et al., 2015; Fung et al., 2022; Jernigan et al., 2022a). It has been reported that 30–46% of
emitted DMS forms HPMTF according to different modeling studies and this falls within the
observational range from NASA Atmospheric Tomography ATom-3 and ATom-4 flight
campaigns where about 30–40% DMS was oxidized to HPMTF along their flight tracks (Fung et
al., 2022; Veres et al., 2020; Novak et al., 2021). Subsequent investigation of the isomerization
rate and heterogeneous loss of HPMTF in cloud droplets and aerosol shows a high production rate
of marine carbonyl sulfide (OCS) from the chemical loss of HPMTF, a potential precursor of
stratospheric sulfate aerosol and significant inhibitor of cloud condensation nuclei (CCN)
formation due to the resulting reduction of surface $SO_2$ (Jernigan et al., 2022a). With the latest
experimental findings on heterogeneous loss process of HPMTF and experimentally validated
oxidation reactions for OCS formation directly from HPMTF it is necessary to include these
reactions as part of the DMS oxidation mechanism as these will have impact on overall yield of
$SO_2$, thus affecting the formation probability of CCN (Jernigan et al., 2022a, b).
**Table 1.** The three DMS oxidation reactions in the standard GEOS-Chem chemical mechanism

| Reactions | Rate constant ($cm^3$ molecule$^{-1}$ s$^{-1}$) | |
|---|---|---|
| $DMS + OH_{(abstraction)} \rightarrow SO_2 + CH_3O_2 + CH_2O$ | $1.20\times10^{-11}exp(-280/T)$ | (R1) |
| $DMS + OH_{(addition)} \rightarrow 0.75\ SO_2 + 0.25\ MSA + CH_3O_2$ | $8.2\times10^{-39}[O_2]exp(5376/T)/(1+1.05\times10^{-5}([O_2]/[M])\ exp(3644/T))$ | (R2) |
| $DMS + NO_3 \rightarrow SO_2 + HNO_3 + CH_3O_2 + CH_2O$ | $1.90\times10^{-13}exp(530/T)$ | (R3) |

Considering these and other consequences of complex DMS oxidation processes, a heavily
simplified oxidation scheme will necessarily neglect potentially important reaction intermediates
along with their production and loss pathways, with implications for the concentration and
distribution of the oxidation products, including particulate sulfate. Differing intermediate
lifetimes further influence sulfur removal and transport depending on the relative dominance of
pathways. Thus, the exclusion of key pathways and intermediate species can lead to errors in the
representation of the spatial distribution of both gas- and particle-phase sulfur species, as well as
global sulfur burden.
The DMS oxidation products sulfate and MSA play an important role in Earth's radiative budget
through cloud droplet formation, and the extent of this role depends on how efficiently they can
produce and grow new particles in the marine atmosphere (Thomas et al., 2010). $SO_2$ can oxidize
in the gas-phase the forming $H_2SO_4$, which acts as a key product contributing to nucleation and
condensational growth as shown in Figure 1. $SO_2$ oxidizing through aqueous chemistry in cloud
droplets does contribute to particle growth rates by providing larger aerosol during cloud
evaporation that acts as more efficient CCN (Kaufman and Tanré, 1994). On the other hand, MSA
might participates in nucleation along with sulfuric acid in presence of amines or ammonia
(Johnson and Jen, 2023). Recent studies have highlighted the importance of aqueous-phase
chemistry in the formation and loss of MSA (Boniface et al., 2000; Chen et al., 2015; Kaufman
and Tanré, 1994; Kulmala et al., 2000).

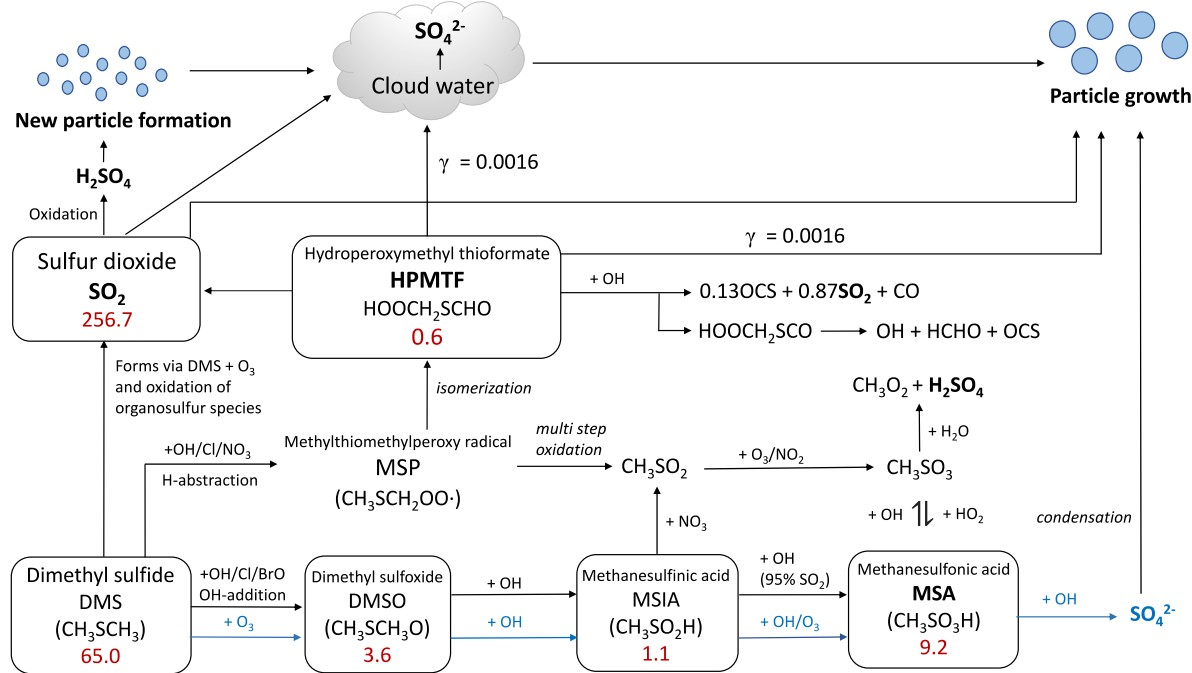

**Figure 1** Modified DMS oxidation mechanism used in this work (simulation MOD) showing the formation of major stable oxidation products (in bold) including the newly identified intermediate HPMTF, and their contribution to new particle formation or growth of existing particles. The blue arrows and text represent aqueous-phase reactions. Numbers inside boxes indicate burden in units of GgS. $\gamma$ values represent reactive uptake co-efficients for heterogeneous loss of HPMTF to cloud and aerosol. Note that $SO_2$ formation from DMS and HPMTF involves multiple oxidation steps in this mechanism, but full pathways are simplified here for visual clarity

Additionally, the recently identified intermediate HPMTF also has the potential for further gas-
phase oxidation. Under cloud-free conditions, HPMTF can undergo gas-phase oxidation by OH,
producing $SO_2$ and eventually leading to the formation of non-sea-salt-$SO_4^{2-}$. This sulfate can
contribute to aerosol formation and growth processes, with climate implications (Galí et al., 2019).
Other work has used direct airborne eddy covariance flux measurements to explain the chemical
fate of HPMTF in the MBL, finding that in cloudy conditions chemical loss due to aqueous phase
reactions in clouds is the major HPMTF removal process (Novak et al., 2021). In the same study,
global model simulations showed a 35% reduction in global annual average $SO_2$ production from
DMS and a 24% reduction in the near-surface (0 to 3 km) global annual average $SO_2$
concentrations over the ocean as a result of this process (Novak et al., 2021). Thus, a complete
representation of cloud loss and aerosol uptake is needed to effectively evaluate the atmospheric
impacts of marine DMS and their connections to cloud formation (Novak et al., 2021; Holmes et
al., 2019).
To better understand the marine sulfur budget, as well as the eventual formation, size distribution,
and seasonality of sulfate aerosol, we use the global chemical transport model GEOS-Chem,
integrating previously developed mechanisms along with newly proposed pathways involving the
formation and loss of the intermediates DMSO, MSIA, and HPMTF. As part of this work, we
further quantify the atmospheric impacts of individual reactions and mechanisms, evaluate
uncertainties in the chemical mechanism, and identify improvements necessary to better represent
the impacts of DMS more accurately on atmospheric chemistry and climate. The resulting

integrated scheme provides a more complete representation of marine sulfur and sulfate aerosol species in marine tropospheric environments compared to the simplified base GEOS-Chem mechanism, with improved comparisons to aircraft and surface observations. Since aerosols are a major contributor to uncertainty in climate forcing, improving oxidation and aerosol formation mechanisms by adding and optimizing neglected reactions in models is a crucial step towards a more mechanistically robust representation of particle yields and sensitivities. We further perform multiple sensitivity tests to investigate how the uncertainty in heterogeneous uptake of the newly identified HPMTF could influence DMS chemistry and tropospheric aerosol formation (Holmes et al., 2019; Novak et al., 2021). In a broader sense our work provides a more detailed story on the heterogeneous loss, fate, and ultimate impacts of DMS and its oxidation products, improving our understanding of a key ocean-atmosphere interaction in the context of global change.

## 2    Methodology

To simulate DMS chemistry and its oxidation products GEOS-Chem global chemical transport model v12.9.3 is used. Impacts on simulated aerosol size, number and mass concentration are considered by coupling the TwO-Moment Aerosol Sectional (TOMAS) aerosol microphysics module with GEOS-Chem v12.9.3 (GC-TOMAS) (https://github.com/geoschem/geos-chem/tree/12.9.3) (Adams and Seinfeld, 2002; Kodros and Pierce, 2017). The default GEOS-Chem chemical mechanism contains detailed $HO_x$–$NO_x$–VOC–$O_3$–halogen tropospheric chemistry along with recently updated halogen chemistry and in-cloud processing (Bey et al., 2001; Holmes et al., 2019; Chen et al., 2017; Parrella et al., 2012; Schmidt et al., 2016; Wang et al., 2019). The DMS emission flux from ocean are controlled by a gas transfer velocity which is dependent on sea surface temperature and wind speed (Johnson, 2010) and a climatology of concentrations in seawater (Lana et al., 2011; Nightingale et al., 2000). The aqueous-phase concentration of $O_3$ in aerosols or cloud droplets is calculated assuming gas-liquid equilibrium and aqueous-phase concentration of OH is calculated following $[OH_{(aq)}] = \delta[OH_{(g)}]$ where, $\delta = 1 \times 10^{-19}$ M cm$^3$ molecule$^{-1}$ (Jacob et al., 2005; Chen et al., 2018).

In this study, TOMAS tracks aerosol number and the mass of each aerosol species in 15 logarithmically sized bins, with sizes in this analysis ranging from 3 nm to 10 μm (Lee and Adams, 2012; Lee et al., 2013). All binned aerosol species undergo interactive microphysics, allowing the calculation of aerosol number budgets (Westervelt et al., 2013). The version of GC-TOMAS used here includes 47 vertical levels, a horizontal resolution of 4º × 5º, and the GEOS-FP data product for meteorological inputs. Simulations are performed for 2018, with 11 months of discarded model spin up. Nucleation is simulated via a ternary nucleation scheme involving water, sulfuric acid, and ammonia with nucleation rates scaled by $10^{-5}$ (Napari et al., 2002; Westervelt et al., 2013). In low-ammonia regions (less than 1 pptv), a binary nucleation scheme involving water and sulfuric acid is instead used (Vehkamäki et al., 2002). Previously GC-TOMAS has been used for aerosol simulations to investigate topics such as the aerosol cloud-albedo effect and cloud condensation nuclei formation (Kodros et al., 2016; Kodros and Pierce, 2017; Pierce and Adams, 2006; Westervelt et al., 2013). Aerosol species available for GC-TOMAS simulations are sulfate, aerosol water, black carbon, organic carbon, mineral dust, and sea salt  (Alexander et al., 2005; Bey et al., 2001; Duncan Fairlie et al., 2007; Pye et al., 2009). The wet and dry deposition scheme for aerosols and gas species are based on previous studies (Amos et al., 2012; Emerson et al., 2020; Liu et al., 2001; Wesely, 1989; Wang et al., 1998).

We refer to simulations performed using only these three DMS oxidation reactions (Table 1) as the "BASE", involving only the direct formation of $SO_2$ and MSA in gas-phase (Chin et al., 1996). We further implement and evaluate a custom chemical mechanism for DMS oxidation, referred to as "MOD" (Table 2-4), representing an integration of three individual DMS oxidation mechanism updates explored previously using GEOS-Chem and CAM6-Chem. This mechanism also includes HPMTF loss to clouds and aerosols via heterogeneous chemistry, dry and wet deposition of HPMTF, along with further improvement based on recent literature updates to chemical kinetics (Chen et al., 2018; Fung et al., 2022; Veres et al., 2020; Novak et al., 2021; Cala et al., 2023). In GC-TOMAS we use specific subroutine that take amount of sulfate produced via in-cloud oxidation and condense it into an existing aerosol size distribution. So, mass of sulfate produced by oxidation is portioned to the various size bins according to the number of particles in that size bin. TOMAS microphysics accounts for $H_2SO_4$ formation based on gas-phase oxidation of $SO_2$

included in the kinetic preprocessor (KPP) equation list valid for the simulation BASE. Since there are additional sources of sulfate in the integrated DMS oxidation mechanism both in gas and aqueous phase, we made necessary changes in the KPP code to explicitly track $H_2SO_4$ formation by gas phase oxidation of $SO_2$. On the other hand, code changes for sulfate formed by heterogeneous oxidation of MSA and HPMTF (in clouds and aerosols) were added in the GEOS-Chem microphysics module that also handles in-cloud oxidation of $SO_2$ in GC version 12.9.3 (Park et al., 2004; Trivitayanurak et al., 2008).

**Table 2.** Overview of the DMS oxidation mechanism via OH-addition pathway.

| Gas-phase reactions | Rate constant ($cm^3$ molecule$^{-1}$ s$^{-1}$) | References |
|---|---|---|
| DMS + OH → DMSO + $HO_2$ | $9.5 \times 10^{-39}[O_2] \exp(5270/T)/(1+ 7.5 \times 10^{-29}[O_2] \exp(5610/T))$ | IUPAC SOx22 (upd. 2006) |
| DMS + BrO → DMSO + Br | $1.50 \times 10^{-14} \exp(1000/T)$ | (Bräuer et al., 2013; Hoffmann et al., 2016) |
| DMS + $O_3$ → $SO_2$ | $1.50 \times 10^{-19}$ | (Du et al., 2007; Burkholder et al., 2020) |
| DMSO + OH → 0.95(MSIA + $CH_3O_2$) | $6.10 \times 10^{-12} \exp(800/T)$ | MCMv3.3.1, (von Glasow and Crutzen, 2004; Burkholder et al., 2020) |
| MSIA + OH → $0.95SO_2$ + $0.95CH_3O_2$ | $9.00 \times 10^{-11}$ | MCMv3.3.1 |
| MSIA + OH → 0.05MSA + $0.05HO_2$ + $0.05H_2O$ | $9.00 \times 10^{-11}$ | (von Glasow and Crutzen, 2004) |
| MSIA + $NO_3$ → $CH_3SO_2$ + $HNO_3$ | $1.00 \times 10^{-13}$ | (von Glasow and Crutzen, 2004; Hoffmann et al., 2016) |

| Aqueous-phase reactions | $k_{298}$ [$M^{-1}s^{-1}$] | References |
|---|---|---|
| DMS (aq) + $O_3$ (aq) → DMSO (aq) + $O_2$ (aq) | $8.61 \times 10^8$ | (Gershenzon et al., 2001; Hoffmann et al., 2016) |
| DMSO (aq) + OH (aq) → MSIA (aq) | $6.65 \times 10^9$ | (Zhu et al., 2003; Hoffmann et al., 2016) |
| MSIA (aq) + OH (aq) → MSA (aq) | $6.00 \times 10^9$ | (Hoffmann et al., 2016; Herrmann et al., 1998) |
| $MSI^-$ (aq) + OH (aq) → MSA (aq) | $1.20 \times 10^{10}$ | (Bardouki et al., 2002; Hoffmann et al., 2016) |
| MSIA (aq) + $O_3$ (aq) → MSA (aq) | $3.50 \times 10^7$ | (Hoffmann et al., 2016; Herrmann et al., 1998) |
| $MSI^-$ (aq) + $O_3$ (aq) → MSA (aq) | $2.00 \times 10^6$ | (Flyunt et al., 2001; Hoffmann et al., 2016) |
| MSA (aq) + OH (aq) → $SO_4^{2-}$ | $1.50 \times 10^7$ | (Hoffmann et al., 2016; Herrmann et al., 1998) |
| $MS^-$ (aq) + OH (aq) → $SO_4^{2-}$ (aq) | $1.29 \times 10^7$ | (Zhu et al., 2003; Hoffmann et al., 2016) |

**Table 3.** Overview of the DMS oxidation mechanism involving HPMTF formation.

| Gas-phase reactions | Rate constant ($cm^3$ molecule$^{-1}$ s$^{-1}$) | References |
|---|---|---|
| MSP ($CH_3SCH_2OO$) → $OOCH_2SCH_2OOH$ | $2.2433 \times 10^{11} \exp(-9.8016e3/T) \times (1.0348 \times 10^8/T^3)$ | (Berndt et al., 2019; Veres et al., 2020; Wollesen de Jonge et al., 2021) |
| $OOCH_2SCH_2OOH$ → HPMTF ($HOOCH_2SCHO$) + OH | $6.0970 \times 10^{11} \exp(-9.489e3/T) \times (1.1028 \times 10^8/T^3)$ | (Berndt et al., 2019; Veres et al., 2020; Wollesen de Jonge et al., 2021) |
| $OOCH_2SCH_2OOH$ + NO → $HOOCH_2S$ + $NO_2$ + HCHO | $4.9 \times 10^{-12} \exp(260/T)$ | MCMv3.3.1 |

| | | |
|---|---|---|
| MSP + HO$_2$ → CH$_3$SCH$_2$OOH + O$_2$ | 1.13×10$^{-13}$exp(1300/T) | MCMv3.3.1, (Wollesen de Jonge et al., 2021) |
| CH$_3$SCH$_2$OOH + OH → CH$_3$SCHO | 7.03×10$^{-11}$ | MCMv3.3.1 |
| CH$_3$SCHO + OH → CH$_3$S + CO | 1.11×10$^{-11}$ | MCMv3.3.1 |
| HPMTF + OH→ HOOCH$_2$SCO + H$_2$O | 4.00×10$^{-12}$ | (Jernigan et al., 2022a) |
| HPMTF + OH→ 0.13OCS + 0.87SO$_2$ + CO | 1.40×10$^{-11}$ | (Jernigan et al., 2022a) |
| OCS + OH → SO$_2$ | 1.13×10$^{-13}$exp(1200/T) | (Jernigan et al., 2022a) |
| HOOCH$_2$SCO → HOOCH$_2$S + CO | 9.2×10$^9$exp(-505.4/T) | (Wu et al., 2015) |
| HOOCH$_2$SCO → OH + HCHO + OCS | 1.6×10$^7$exp(-1468.6/T) | (Wu et al., 2015) |
| HOOCH$_2$S + O$_3$ → HOOCH$_2$SO + O$_2$ | 1.15×10$^{-12}$exp(430/T) | (Wu et al., 2015) |
| HOOCH$_2$S + NO$_2$ → HOOCH$_2$SO + NO | 6.0×10$^{-11}$exp(240/T) | (Wu et al., 2015) |
| HOOCH$_2$SO + O$_3$ → SO$_2$ + HCHO + OH + O$_2$ | 4.0×10$^{-13}$ | (Wu et al., 2015) |
| HOOCH$_2$SO + NO$_2$ → SO$_2$ + HCHO + OH + NO | 1.2×10$^{-11}$ | (Wu et al., 2015) |

**Table 4.** Overview of the MSA-producing branch of the H-abstraction pathway of DMS oxidation.

| Gas-phase reactions | Rate constant (cm$^3$ molecule$^{-1}$ s$^{-1}$) | References |
|---|---|---|
| DMS + OH → MSP (CH$_3$SCH$_2$OO) + H$_2$O | 1.12×10$^{-11}$exp(-250/T) | IUPAC SOx22 (upd. 2006) |
| DMS + Cl → 0.45MSP + 0.55C$_2$H$_6$SCl + 0.45HCl | 3.60×10$^{-10}$ | (Fung et al., 2022; Enami et al., 2004) |
| C$_2$H$_6$SCl → DMSO + ClO | 4.00×10$^{-18}$ | (Hoffmann et al., 2016; Urbanski and Wine, 1999) |
| DMS + NO$_3$ → MSP + HNO$_3$ | 1.9×10$^{-13}$exp(520/T) | MCMv3.3.1, (Novak et al., 2021; Wollesen de Jonge et al., 2021; Atkinson et al., 2004) |
| MSP + NO → CH$_3$SCH$_2$(O) + NO$_2$ | 4.9×10$^{-12}$exp(260/T) | MCMv3.3.1 |
| MSP + MSP → 2HCHO + 2CH$_3$S | 1.00×10$^{-11}$ | (von Glasow and Crutzen, 2004) |
| CH$_3$SCH$_2$(O) → CH$_3$S + HCHO | 1.0×10$^6$ | MCMv3.3.1 |
| CH$_3$S + O$_3$ → CH$_3$S(O) | 1.15×10$^{-12}$exp(430/T) | MCMv3.3.1; (Atkinson et al., 2004) |
| CH$_3$S + O$_2$ → CH$_3$S(OO) | 1.20×10$^{-16}$exp(1580/T) | MCMv3.3.1; (Atkinson et al., 2004) |
| CH$_3$S + NO$_2$ → CH$_3$SO + NO | 3.00×10$^{-12}$exp(210/T) | IUPAC SOx60 (upd. 2006); (Atkinson et al., 2004) |
| CH$_3$S(O) + O$_3$ → CH$_3$(O$_2$) + SO$_2$ | 4.00×10$^{-13}$ | IUPAC SOx61 (upd. 2006); (Borissenko et al., 2003) |
| CH$_3$SO + NO$_2$ → 0.75CH$_3$SO$_2$ + 0.75NO + 0.25SO$_2$ + 0.25CH$_3$O$_2$ + 0.25NO | 1.20×10$^{-11}$ | (Borissenko et al., 2003; Atkinson et al., 2004) |
| CH$_3$S(OO) → CH$_3$(O$_2$) + SO$_2$ | 5.60×10$^{16}$exp(-10870/T) | (Atkinson et al., 2004) |
| CH$_3$S(OO) → CH$_3$SO$_2$ | 1.00 | (Campolongo et al., 1999; Hoffmann et al., 2016) |
| CH$_3$S(OO) → CH$_3$S + O$_2$ | 3.50×10$^{10}$exp(-3560/T) | MCMv3.3.1 |
| CH$_3$SO$_2$ + O$_3$ → CH$_3$SO$_3$ + O$_2$ | 3.00×10$^{-13}$ | MCMv3.3.1; (von Glasow and Crutzen, 2004) |
| CH$_3$SO$_2$ → CH$_3$(O$_2$) + SO$_2$ | 5.00×10$^{13}$exp(-9673/T) | MCMv3.3.1; (Barone et al., 1995) |

| | | | |
|---|---|---|---|
| $CH_3SO_2 + NO_2 \rightarrow CH_3SO_3 + NO$ | $2.20 \times 10^{-11}$ | | (Atkinson et al., 2004) |
| $CH_3SO_3 + HO_2 \rightarrow MSA$ | $5.00 \times 10^{-11}$ | | MCMv3.3.1; (von Glasow and Crutzen, 2004) |
| $CH_3SO_3 \rightarrow CH_3(O_2) + H_2SO_4$ | $5.00 \times 10^{13} exp(-9946/T)$ | | MCMv3.3.1 |
| $MSA + OH \rightarrow CH_3SO_3$ | $2.24 \times 10^{-14}$ | | MCMv3.3.1 |

To examine the sensitivities of size-resolved aerosol formation and growth to DMS chemistry
modifications, model simulations are conducted as summarized in Table 5. Output from
simulations MOD and MOD_noHetLossHPMTF was then compared against simulation BASE to
understand the contribution of these additional chemical reactions on spatial pattern of the surface
concentration of major oxidation products of DMS.
**Table 5.** List of mechanisms used in GEOS-Chem-TOMAS simulations.

| Model Runs | Mechanism | HPMTF Cloud Loss[*] | HPMTF Aerosol Loss[*] |
|---|---|---|---|
| BASE | All reactions from Table 1 | - | - |
| MOD_noHetLossHPMTF | All reactions from Table 2-4 | Off | Off |
| MOD | All reactions from Table 2-4 | On | On |

[*] Instantaneous formation of sulfate via HPMTF cloud and aerosol loss uses a reactive uptake co-efficient ($\gamma$) of
224 0.0016.

As shown in Table 2, the modified DMS chemistry simulations examined here include gas- and
aqueous-phase oxidation of DMS and its intermediate oxidation products by OH, $NO_3$, $O_3$, and
halogenated species as previously explored in an older version of GEOS-Chem (Chen et al., 2018).
The aqueous-phase reactions in cloud droplets and aerosols were parameterized assuming a first-
order loss of the gas-phase sulfur species (Chen et al., 2018). Further building upon this previous
mechanism, the scheme used here also includes the formation and loss of HPMTF as previously
tested in the global climate model CAM6-Chem as shown in Table 3 (Veres et al., 2020). Table 4
presents the third piece of the mechanism: a gas-phase MSA-producing branch of the H-abstraction
pathway in the DMS chemistry bridging the other two sets of the reactions (Fung et al., 2022). To
avoid addition of $SO_3$ oxidation chemistry we have replaced $SO_3$ with $H_2SO_4$ followed by previous
work for the decomposition reaction of $CH_3SO_3$ (Table 4). A similarly integrated mechanism
(Table 2-4) has been previously explored using the CAM6-Chem model with a focus on radiation
budget impacts, which is improved in this work through updates rate constants and the inclusion
of additional relevant reactions (Fung et al., 2022; Novak et al., 2021; Wollesen de Jonge et al.,
2021; Cala et al., 2023). The newly added reactions and their respective rate constants are largely
based on the MCMv3.3.1 and the literature cited in the Table 2-4 reference list. We use a rate
constant of $1.40 \times 10^{-11}$ cm$^3$ molecules$^{-1}$s$^{-1}$ for HPMTF + OH, which was previously determined
based concentrations of other known sulfur species (DMS, DMSO, $SO_2$ and methyl thioformate;
MTF; $CH_3SCHO$; a structurally similar proxy to HPMTF) and evaluated by box model (Jernigan
et al., 2022a). An exploration of reaction rate uncertainty for the HPMTF+OH reaction (Table 3),
including both high and low end limits of $5.5 \times 10^{-11}$ cm$^3$ molecules$^{-1}$s$^{-1}$ and $1.4 \times$
$10^{-12}$ cm$^3$ molecules$^{-1}$s$^{-1}$ resulted in only minor impacts on the fate of HPMTF and ultimate sulfate
formation in our simulations (Novak et al., 2021; Wu et al., 2015).
Model sensitivity simulations were also performed with (case "MOD") and without HPMTF
heterogeneous uptake to clouds and aerosols (case "MOD_noHetLossHPMTF") to account for
how much of the DMS-derived HPMTF eventually forms $SO_2$ in the presence of these additional
loss processes (Table 5). Previous work shows that aerosol surface chemistry causes additional
decreases in HPMTF mixing ratios, primarily over land, and that the loss of HPMTF in clouds is
larger (36%) than losses from aerosols (15%) when using an uptake coefficient of $\gamma = 0.01$ for both
processes (Novak et al., 2021). In this work, based on recent laboratory measurements, we use a
smaller uptake coefficient ($\gamma = 0.0016$) for HPMTF loss to aerosols and clouds (Table 5) (Jernigan
et al., 2022b). We assume HPMTF directly produces sulfate in cloud and aerosol followed but
previous work even though there is uncertainty in the fate of HPMTF heterogeneous loss (Zhang
and Millero, 1993; Novak et al., 2021; Jernigan et al., 2022a). For the aqueous-phase reactions
listed in Table 2, including the oxidation of intermediates DMSO and MSIA in cloud droplets and
aerosols, a first-order loss of the gas-phase sulfur species was assumed following previously used
parameterizations and physical parameter values (Chen et al., 2018). Alongside the gas-phase and
aqueous-phase reactions relevant to the added DMS oxidation mechanism contributing to the
formation of $SO_2$ and sulfate, the default version of GC-TOMAS used here also includes in-cloud
oxidation of $SO_2$ by $H_2O_2$, $O_3$, and $O_2$ catalyzed by transition metals (Mn, Fe), as well as the loss
of dissolved $SO_2$ by HOBr and HOCl, all of which are passed to TOMAS to account for sulfate
production (Chen et al., 2017; Wang et al., 2021).
All simulations are conducted for the year 2018, which was chosen to match the model simulation
with the dates of the NASA Atmospheric Tomography flight campaign (ATom-4) offering
observational data for HPMTF, DMS and $SO_2$. Rate coefficients for all gas-phase sulfur reactions
are obtained from the most recent JPL report and other references while sulfur product yields for
gas-phase reactions are obtained from various laboratory and modeling studies (Burkholder et al.,
2020; Lucas and Prinn, 2002; Hoffmann et al., 2016; Gershenzon et al., 2001; Kowalczuk et al.,
2003; Zhou et al., 2019; Jernigan et al., 2022a). The simulations included sea salt debromination
except for some sensitivity tests described below (Zhu et al., 2019; Schmidt et al., 2016). In all our
simulations including MOD, DMS is advected and undergoes chemical loss and transport but does
not undergo dry or wet deposition. However, dry and wet deposition of oxidation products such as
DMSO, MSIA, MSA and HPMTF are included.
We note that previous work has explored the impact of MSA on aerosol growth, including
modifications within TOMAS to represent this process (Hodshire et al., 2019). We do not include
this process here. Future work is recommended to examine its importance in the context of the
chemistry updates presented here.

## 3 Result and discussion

3.1 Model-Observations Comparison

3.1.1 Surface DMS mixing ratio

We compared the modeled DMS mixing ratio averaged for each month with the observational data collected at Crete Island (35° N, 26° E) and Amsterdam Island (37° S, 77° E) (Kouvarakis and Mihalopoulos, 2002; Chen et al., 2018; Castebrunet et al., 2009). Comparing simulations BASE and MOD, we find a closer match with DMS observations for simulations using modified DMS chemistry for both observation data shown in Figure 2. Modeled DMS mixing ratios calculated using base chemistry show strong positive bias during the months of May and June for Crete Island. By comparison, during the same period the modeled DMS mixing ratios calculated with modified chemistry reduces the bias from 102% to 42%. Similarly, for Amsterdam Island major overpredictions are apparent for the BASE simulation compared to MOD for the months of May-August. One reaction that may play a role in this shift is DMS + BrO, which as indicated earlier is responsible for a faster overall chemical loss of DMS, in particular over the southern hemisphere high latitudes. Beside DMS chemistry, sea surface DMS concentration is also proven to affect the modeled DMS mixing ratio (Chen et al., 2018). But the aim of this study is to investigate the chemistry aspect of DMS oxidation, so we did not explored how change in DMS seawater climatology and thus their emission influence the surface DMS mixing ratio.

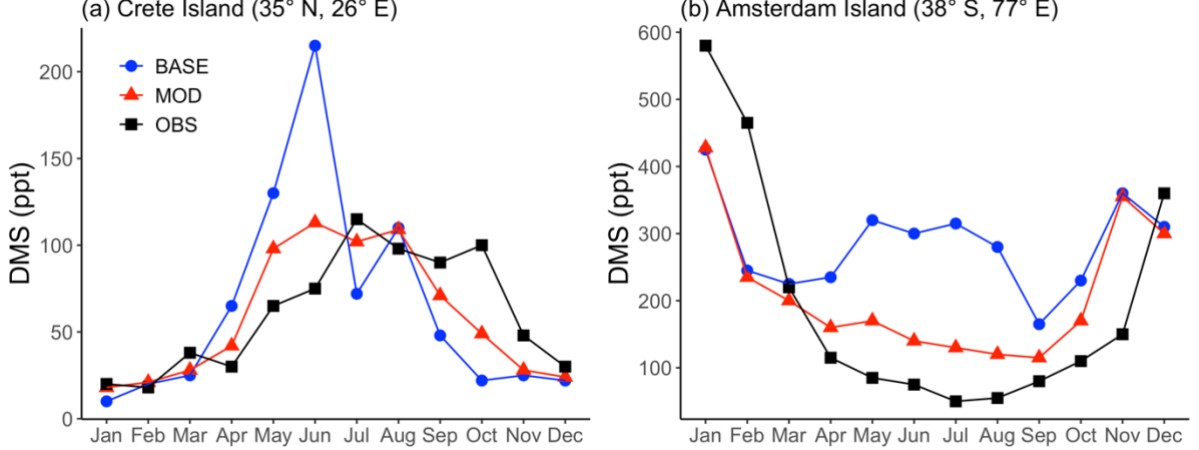

**Figure 2** Observed (OBS) monthly mean surface DMS mixing ratios at (a) Crete Island and (b) Amsterdam Island compared with simulations BASE and MOD. Simulations are described in Table 5.

3.1.2 Comparison with aircraft observations

We further evaluate model output through a comparison with ATom-4 aircraft observations for specific days of measurement for DMS, HPMTF and $SO_2$ as shown in Figure 5. For this comparison, the model is sampled at the time and location of aircraft measurements by ATom-4 using the planeflight diagnostic of GEOS-Chem.

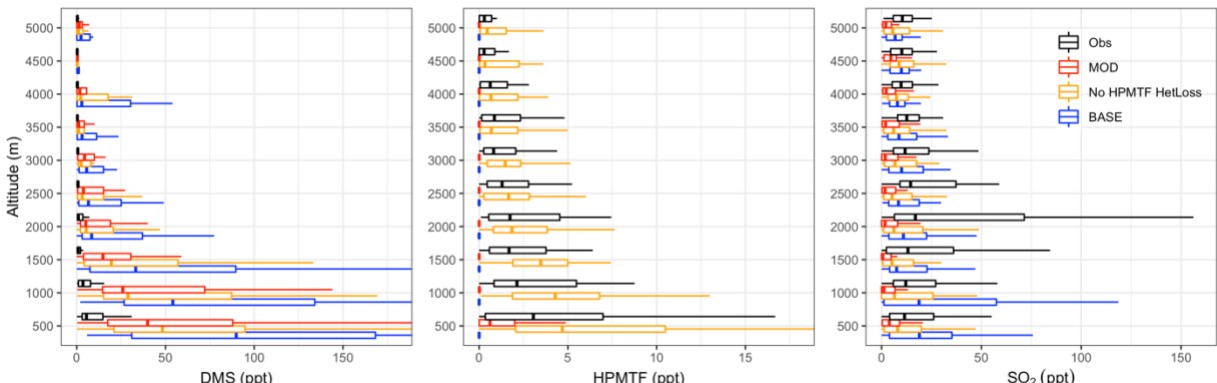

**Figure 3** Vertical profiles of (a) DMS, (b) HPMTF and (c) $SO_2$ mixing ratios from ATom-4 observations (black) and model with simulation MOD sampled along the ATom-4 flight tracks (red) binned every 500 m of flight altitude. Also shown are modeled results without HPMTF heterogeneous loss with simulation MOD_noHetLossHPMTF (yellow), and for BASE GEOS-Chem chemistry (blue). Box plot whiskers show full range of distribution at each altitude bin. DMS observations are from Whole Air Samples (WAS) while HPMTF DC-8 observations are from iodide ion chemical ionization time-of-flight mass spectrometer (CIMS). $SO_2$ observations from ATom-4 campaign were measured by Laser Induced Fluorescence (LIF).

DMS concentrations measured during ATom-4 by whole air sampler (WAS) and modified
chemistry simulation values for nearest neighbor grid cells are shown in Figure 3a across different
altitude. In general, the modeled DMS concentrations are significantly higher than those observed
during ATom-4 missions especially close to the surface. However, model DMS concentrations
decrease more rapidly than the measurement with altitudes indicating vertical mixing could be one
of the underlying reasons for this trend. Even with this near surface bias, simulation MOD relative
to BASE has greater DMS losses and a shorter DMS lifetime (from 1.5 d to 0.9 d) reducing the
gap between modeled and observed concentration compared to simulation BASE. The reduction
in modeled DMS is largest over the Southern Ocean (shown later in Fig. 5b) where oxidation by
BrO and $O_3$ in the aqueous phase plays the major role in reducing DMS concentration, thereby
reducing the model-observation bias (Fig. 3b). Remaining model biases could be at least partially
attributed to model uncertainty in oxidant concentrations and cloud cover. The heterogeneous loss
of HPMTF has minimal impact on DMS concentration and its vertical profile.
For HPMTF, Figure 3b shows that the observed and modeled HPMTF concentrations remain
largely below 15 ppt. Agreement between observations and modeled HPMTF mixing ratios in the
vertical profile (Fig. 3b) is poor for simulation MOD even close to the surface. Removing all
heterogeneous loss of HPMTF improves model comparisons aloft, though surface concentrations
become overestimated (yellow line of Fig. 3b), showing a high sensitivity to cloud and aerosol
loss processes. We also find that the modeled HPMTF:DMS ratios range from 0.15:1 to 0.5:1 on
a daily basis in most cases for when there is no heterogeneous loss of HPMTF, compared to 0.5:1
observed during ATom-4 using the calibration maintained during measurement, implying
reasonably good agreement for this value over daily time scales (Veres et al., 2020). The SARP
flight campaign data has reported much lower HPMTF:DMS ratios (< 0.2) on cloudy days which
is relatable to modeled HPMTF with simulation MOD (Novak et al., 2021). For simulation MOD,
the modeled HPMTF:DMS ratio is 0.03:1 for until 0.5 km and then approaches zero with
increasing altitude, indicating the need for additional work to better constrain production and loss
processes of this intermediate. Our simulations indicate that cloud loss is the dominant modeled
removal process of HPMTF, consistent with previous findings, while gas-phase OH oxidation
plays a minor role (Novak et al., 2021). Thus, the addition of cloud uptake dramatically decreases
HPMTF concentrations throughout the troposphere. Overall, this allows only 10% of HPMTF
produced to end up as $SO_2$ with about 89% lost to clouds and aerosol and thus removed from the
system, resulting net reduction in mean global $SO_2$ by about 40% along with other chemical
processes involved for this reduction as well. Previous work focusing entirely on gas-phase and
heterogeneous loss of HPMTF shows a much higher bias for both DMS and HPMTF during cloudy
and clear sky conditions using the same model and a condensed DMS oxidation mechanism,
indicating that the addition of gas-phase and heterogeneous oxidation of DMS including additional
intermediates such as DMSO and MSIA further reduce model biases for HPMTF with remaining
overestimation of the multiphase loss for HPMTF (Novak et al., 2021).
We also compared the $SO_2$ concentrations measured during ATom-4 by Laser Induced
Fluorescence (LIF) and simulation MOD values for nearest neighbor grid cells are shown in Figure
3c across different altitude. Modeled surface $SO_2$ concentrations are lower than those observed
during ATom-4 missions across the vertical scale shown here for simulation MOD. The greater
$SO_2$ losses results in a shorter $SO_2$ lifetime (from 1.4 d to 1.3 d) for simulation MOD relative to
simulation BASE. The reduction in modeled $SO_2$ is largest over the Southern Ocean (shown later
in Fig. 7a) where heterogeneous oxidation of HPMTF is most efficient and irreversible. Besides,
the OH addition channel of DMS does not directly produce $SO_2$ causing further reduction in the
concentration relative to BASE. Removing the heterogeneous loss of HPMTF increases the
modeled $SO_2$ compared to simulation MOD with underprediction remaining. Remaining model
biases could be at least partially attributed to uncertainty in DMS oxidation processes along with
other non-DMS sources contributing high concentration of $SO_2$. Aside from uncertainty in DMS
emissions and oxidation, recent understanding of marine sulfur chemistry such as methanethiol
($CH_3SH$) oxidation has been reported as an significant source of $SO_2$ in the marine atmosphere
and could help reduce the bias, a possibility deserving further investigation (Berndt et al., 2023;
Novak et al., 2022). Overall the DMS oxidation chemistry implemented in this work reduces the
model observation bias close to the surface (up to 1km) compared to BASE GEOS-Chem
chemistry.
Besides the vertical profile shown in Figure 3b, the global mean surface mixing ratio of HPMTF
with simulation MOD_noHetLossHPMTF for May 2018 is plotted in Figure 4 and compared with
the observational measurement of HPMTF made during the ATom-4 mission during the NASA
DC-8 flight campaign, which sampled the daytime remote marine atmosphere over the Pacific and
Atlantic Oceans. The ATom-4 measurements were carried out during daytime hours between April
24 and May 21, 2018 for 21 non-continuous days.
For this campaign, flight patterns covered vertical profiles from 0.2 to 14 km above the ocean
surface. The flight leg duration was 5 minutes and boundary layer altitude of 150 to 200 m above
the ocean surface. Since most of these measurement days are within the month of May 2018, here
we compare observations with modeled output of mean surface concentration of HPMTF for this
month. With the rate of isomerization reaction used in previous work, we find spatial patterns of
monthly mean surface concentrations are generally well captured (Jernigan et al., 2022a). Overall,
we find that the simulation MOD_noHetLossHPMTF results in better agreement with existing
overprediction for the vertical profile (Fig. 3b) and global surface layer HPMTF levels (Fig. 4)
compared to previous modeled approaches using the CAM-chem model (Veres et al., 2020).

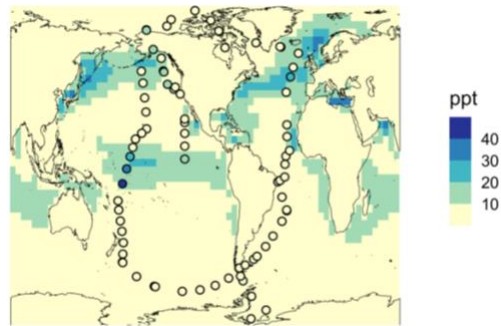

**Figure 4** Geographic distribution of May 2018 monthly mean surface-layer mixing ratio of HPMTF for simulation MOD_noHetLossHPMTF mechanism represented for May 2018. The circles represent measurements of HPMTF during the ATom-4 mission by NASA DC-8 flight tracks with a limit of detection <1 ppt.


3.2 DMS burden and oxidation pathways
We find that the global burden of DMS in the MOD simulation is 65 Gg S (Table B1), 40% lower
than what we find with the simulation BASE (108 Gg S). Even with this 42% reduction, global
burdens are still well within the range of 9.6–150 Gg S suggested in other studies (Faloona, 2009;
Kloster et al., 2006). Figure 5a shows that surface DMS mixing ratios are highest in the North
Pacific and North Atlantic oceans for June-July-August (JJA) and in the Southern Ocean during
the months of December-January-February (DJF), revealing the underlying seasonality of DMS
emissions. According to previous studies, the highest DMS concentrations usually occur in
summer months due to higher rates of primary production in the presence of adequate solar
irradiation and high temperatures for both hemisphere (Galí et al., 2018; Lana et al., 2011; Wang
et al., 2020). In simulation MOD, the global mean surface-layer DMS burden was higher in SH
for DJF and lower in NH for JJA which is due to larger ocean area in the SH than NH. We also
find that the reactions of this expanded DMS oxidation mechanism collectively contribute to
reductions in mean surface-layer DMS concentration of 58% and 22% compared to BASE for JJA
and DJF respectively (Fig. 5b). These reductions are due primarily to the addition of multiple new
chemical loss pathways compared to BASE, which are especially impactful during JJA months
due to due to elevated BrO in the SH winter and also higher $O_3$ and OH concentration in the NH
summer compared to the SH summer (Zhang et al., 2018; Pound et al., 2020).

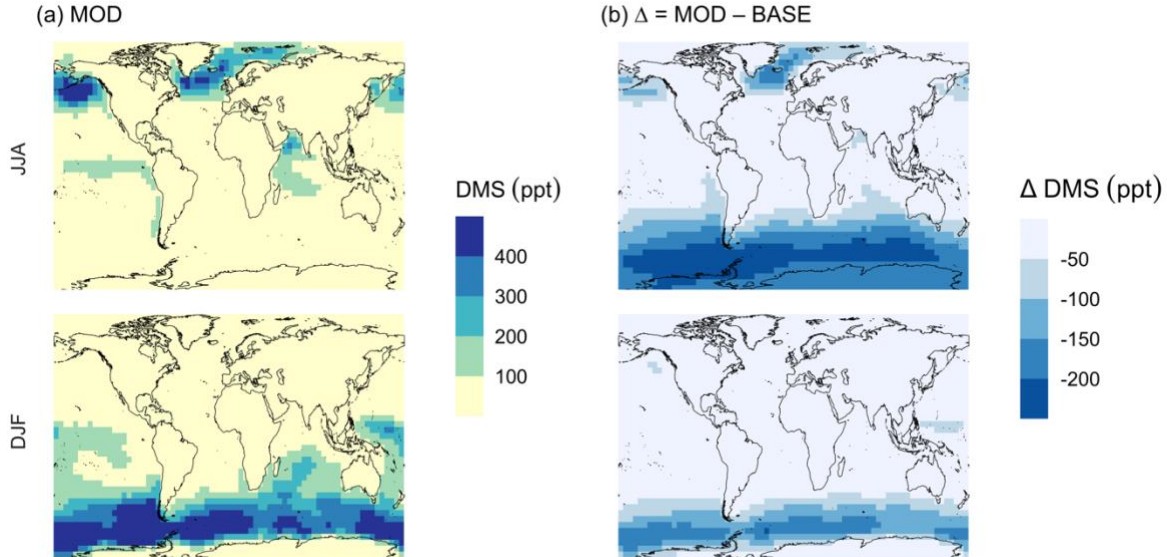

**Figure 5** Geographic distribution of mean surface DMS mixing ratio (ppt) for simulation (a) MOD and (b) difference between simulations from its baseline, $\Delta$ = MOD − BASE from GEOS-Chem simulations. Here, JJA and DJF represent June-July-August and December-January-February respectively. Simulations are described in Table 5.

As shown in Fig. 5b, this DJF DMS reduction is seen mainly over the Southern Ocean and is
largely attributable to faster chemical losses through the added reactions of DMS + BrO and
$DMS_{(aq)} + O_{3(aq)}$, which in earlier work was hypothesized as a possible reason for high model biases
in the absence of detailed halogen chemistry (Chen et al., 2016). The global lifetime of DMS
decreases from 1.5 days in the BASE simulation to 0.9 day in the MOD simulation.
These values are comparable to the range of 0.8–2.1 d reported by previous studies (Chen et al.,
2018; Fung et al., 2022). The global DMS emission flux ($F_{DMS}$) from ocean to the atmosphere is
22 Tg S yr$^{-1}$ and is within the range of 11– 28 Tg S yr$^{-1}$ simulated by GEOS-Chem and other
models in previous studies (Lennartz et al., 2015; Fung et al., 2022; Chen et al., 2018; Hezel et al.,
2011; Spracklen et al., 2005). Our $F_{DMS}$ is higher than the 18 Tg S yr$^{-1}$ which uses sea surface
DMS concentration from Kettle et al. (1999) as reported (Chen et al., 2018) indicating the DMS
emission varies with change in sea surface DMS climatology. The analysis and improvement of
DMS emissions directly is not a part of this work, but we note that improved and validated
inventories for DMS will certainly play a role in subsequent oxidation product comparisons. We
recommend ongoing evaluation of DMS emissions inputs to complement the expanded chemical
mechanism development we present here.
In the BASE simulation the chemical loss of DMS acts as its only sink (as opposed to dry and wet
deposition), leading to a full conversion yield of DMS into $SO_2$ (82.5%) and MSA (17.5%) (Fig.
A3a). Figure 6 shows that in simulation MOD with updated DMS oxidation scheme DMS is
mainly oxidized by OH in the gas phase, with 27.6% of losses proceeding via the H-abstraction
channel and 38.6% via the OH-addition pathway, together contributing up to 66.2% of global
average loss with high regional contribution over the tropical oceans via the abstraction channel
where surface OH is the highest. $NO_3$ oxidation of DMS accounts for another 11.2% of global
DMS chemical losses, comparable to values found in previous studies (Chen et al., 2018; Fung et
al., 2022). Over the ocean, the $NO_3$ loss pathway is strongest in the NH coastal regions due to
outflow of $NO_x$ sources from over the land whereas for the SH values are generally less than 10%.
Oxidation by BrO is responsible for 18.4% of the global DMS removal, falling within the
previously estimated range of 8%–29% (Boucher et al., 2003; Khan et al., 2016; Chen et al., 2018).
Regionally, its contribution can reach 50%–60% over high latitudes of the Southern Hemisphere
as well as to the north near the Arctic Ocean, consistent with previous box model studies based on
the availability of high BrO and low OH and $NO_3$ for those regions (Hoffmann et al., 2016). DMS
+ $O_3$ accounts for 2.2% (aqueous) and 0.9% (gas phase) of global surface DMS loss. The higher
contribution from BrO and lower from $O_3$ using this mechanism compared to some previous
studies could be explained in part by the recently implemented sea-salt debromination mechanism
in GEOS-Chem, resulting in a much higher background level of BrO as well as lower $O_3$
abundance, especially in the southern hemisphere (Boucher et al., 2003; Chen et al., 2018; Fung
et al., 2022; Sherwen et al., 2016a; Wang et al., 2021). To further quantify the importance of the
sea salt debromination mechanism, we perform an emissions sensitivity test by turning this
emission source off while using updated MOD chemistry (Fig. A1). As would be expected, these
simulations show much lower BrO formation (as shown in Fig. A6) and resulting chemical
impacts, with overall oxidation contributions comparable to previous literature (Schmidt et al.,
2016; Wang et al., 2021). We find that under this scenario the relative contribution of BrO for
DMS loss decreases to 2.2%, while the DMS + $O_3$ pathway increases to 43.3% (aqueous) and 1.4%
(gas phase), and the DMS + OH pathway increases to 31.0% (abstraction) and 48.0% (addition)
of global surface DMS loss (Fig. A1). The DMS loss via interaction with $NO_3$ also increases to
2.0% when sea salt debromination is turned off in the mechanism. The relative contributions of
other oxidants remain mostly unaffected in the BrO sensitivity test.

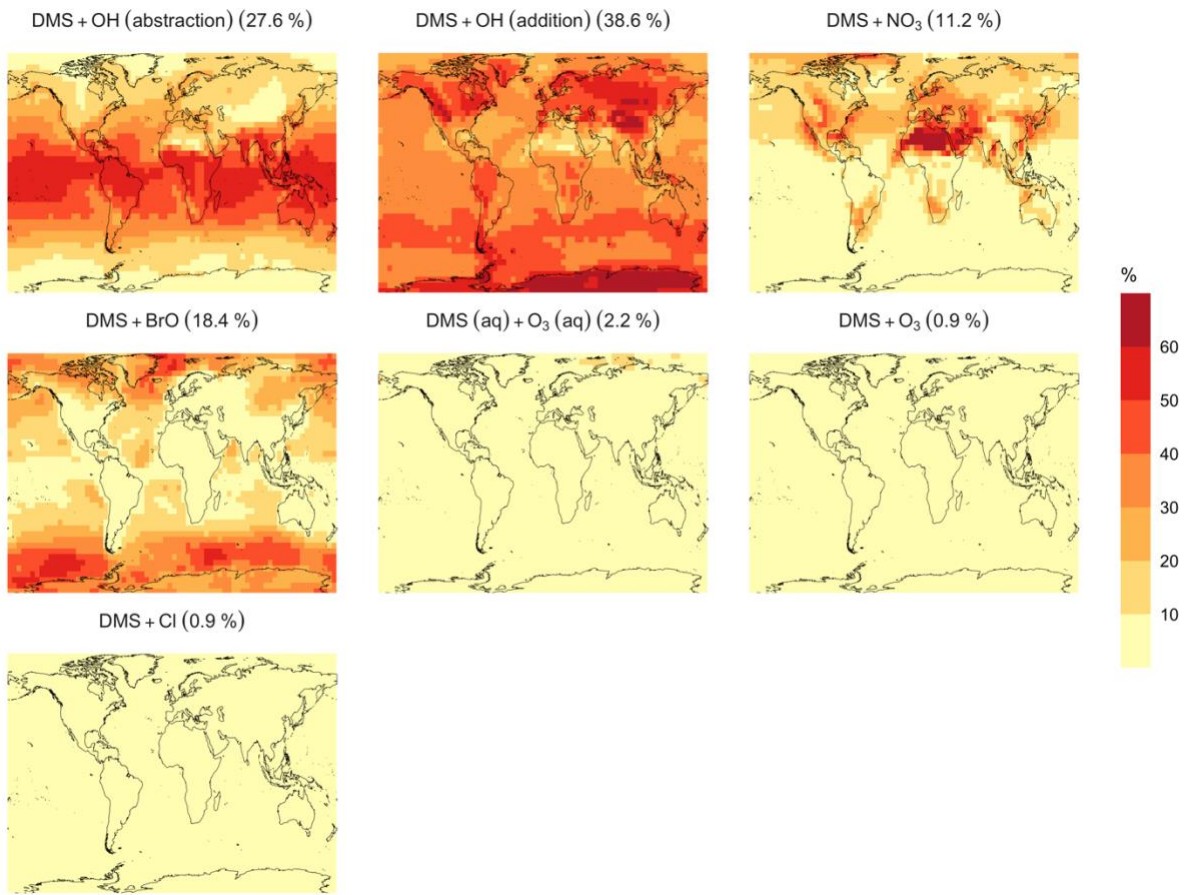

**Figure 6** Geographic distribution of the annual mean surface layer fraction of total DMS oxidation (percent) attributed to different tropospheric oxidants for simulation MOD (described in Table 5). Percentages in parentheses indicate the average contribution to global chemical loss for the fraction of DMS emitted for each reaction pathways presented here.

Regionally, the fractional contribution of aqueous-phase DMS + $O_3$ to DMS oxidation can be up
to 10%–20% over high-latitude oceans especially with the sea salt debromination is turned off
(Fig. A1), which is in the middle of the 5%–30% contribution to high-latitude DMS losses
previously reported (Chen et al., 2018; Fung et al., 2022; von Glasow and Crutzen, 2004). The Cl
oxidation reaction contribute about 0.9% for with and without sea salt debromination to the
chemical removal of DMS, consistent with some previous studies (Atkinson et al., 2004; Fung et
al., 2022). This does differ from other reported values however, including those from a global
model study (4%) and box model simulations (8% –18%) (Chen et al., 2018; Hoffmann et al.,
2016; von Glasow and Crutzen, 2004). It's worth noting that none of the studies reporting such
high Cl contributions included HPMTF formation and loss. Ongoing uncertainties associated with
model-observation bias of Cl should be further resolved to get better representation of halogenated
species contributions to DMS loss (Wang et al., 2021). Due to slower reaction kinetics and lower
fractional contribution reported earlier compared to BrO with DMS and uncertainty in surface
concentration and kinetics for photochemically generated halogenated species such as Br, IO we
did not include them in our chemical scheme (Chen et al., 2018).
3.3 Implications of the extended DMS oxidation mechanism
Figure 7 shows that the MOD simulation results in 40% reduction of surface layer $SO_2$ relative to
BASE, but a huge increase in $SO_4^{2-}$ in most regions. These changes suggest that the combination
of gas-phase and aqueous-phase reactions results in a higher net yield of MSA and HPMTF and a
lower net yield of gas-phase $SO_2$. Additionally, comparison of simulation MOD relative to
MOD_noHetLossHPMTF (Fig. A2a) shows that loss of HPMTF in cloud droplets and aerosol
reduces the global mean production of $SO_2$ by 21.4%, contributing to the $SO_2$ reduction and
increasing mean surface layer sulfate by 12.4% (Fig. A2b). This reduction in $SO_2$ is expected to
reduce the availability of gas-phase sulfuric acid for new particle formation by nucleation (Clarke
et al., 1998a). Total $SO_4^{2-}$ increases over the ocean, however, because the increased $SO_4^{2-}$
production from rapid loss of MSA and HPMTF in aqueous-phase offsets the reduced oxidation
of $SO_2$ (Fig. 7b). In addition to that, reduced gas-phase sulfur species such as $CH_3SO_3$ also
contribute to sulfate formation in our mechanism as followed by other works (Fung et al., 2022).

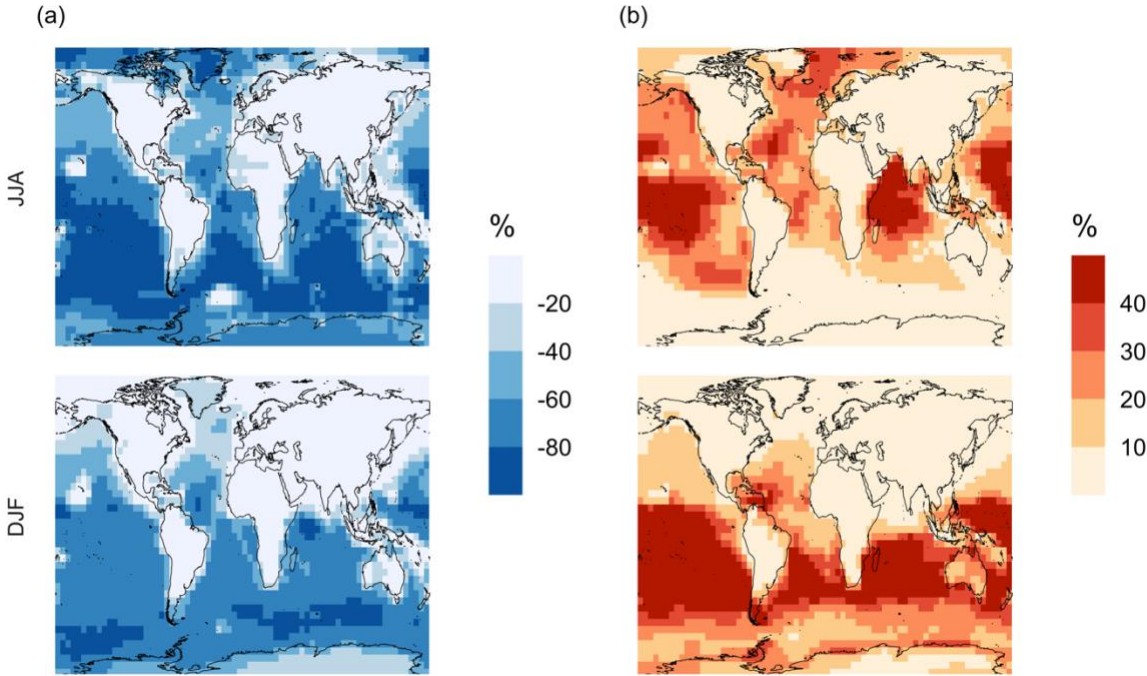

**Figure 7** Percent change in simulated surface layer (a) $SO_2$ and (b) $SO_4^{2-}$ for simulation MOD relative to BASE for June, July and August mean (JJA) and December, January, and February mean (DJF). Simulations are described in Table 5.

Qualitatively, the regions showing the highest percent changes of $SO_2$ are consistent with previous
studies that included HPMTF chemistry and loss processes though the extent of this reduction is
much higher with the integrated mechanism used in our study (Fig. 7a) (Novak et al., 2021). The
regions with the largest percent change in $SO_2$ reduction are those where DMS oxidation
contributes most to $SO_2$, and where HPMTF production and in-cloud oxidation of HPMTF are
efficient. This spatial pattern thus helps us to identify where the production and heterogeneous loss
of HPMTF and MSA is enhanced. One of the reactions that possibly contributes to delayed
formation and reduction of $SO_2$ concentration is the first-generation OCS formation from OH
oxidation of HPMTF. We find that addition of cloud and aerosol loss significantly decreases the
OCS production, especially in high cloud cover regions as previously reported (Jernigan et al.,
2022a). Even though the cloud loss of HPMTF increases the production of surface sulfate, the total
global sulfate burden we calculate increases by only 6.5% from the BASE sulfate burden of around
575 Gg S. This can be attributed to minor contribution of DMS and its intermediate oxidation
products in $SO_2$ production compared to other non-DMS derived sources. In addition, the
production of stable intermediate oxidation products delay the conversion of $SO_2$ to $SO_4^{2-}$ and
modify its spatial distribution in the marine environment. Thus, we should expect these aqueous
phase oxidation products to contribute to particle mass rather than increase the number of nucleated
particles, as suggested in other studies  (Clarke et al., 1998b; Novak et al., 2021; Williamson et al.,
490  2019).

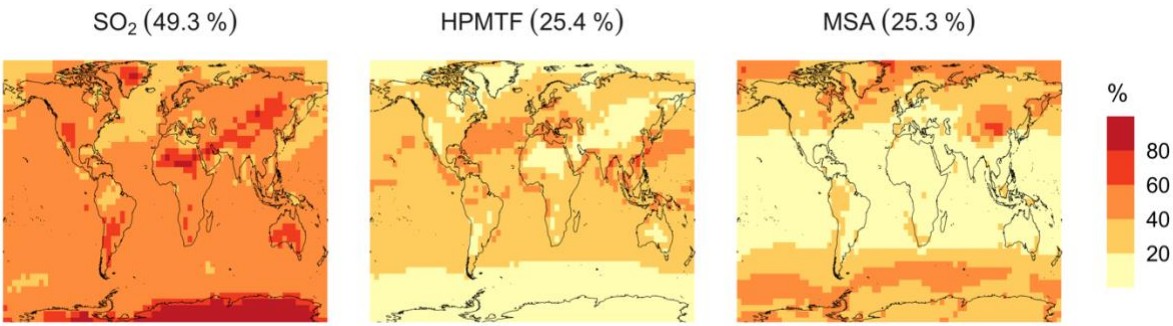

**Figure 8** Simulated branching ratio (in %) of the DMS oxidation mechanism considering $SO_2$, HPMTF and MSA
as major terminal oxidation products calculated from their annual total production rate for simulation MOD.

The spatial distribution of product branching ratios of DMS oxidation is shown in Figure 8. Here,
25.4% of the annual total DMS oxidation will end up as HPMTF, while final $SO_2$ yield decreases
to 49.3% compared to 82.5% for the BASE simulation (Fig. A3a). The terminal HPMTF branch
represents sulfur removed from the system by cloud and aerosol uptake of HPMTF, leading to a
reduced overall formation of $SO_2$. With sea salt debromination turned off, modified chemistry
forms even more HPMTF (27.7%), slightly higher $SO_2$ (51.3%), and lowers the yield of MSA to
21.0% (25.3% with the sea salt debromination on), underscoring the importance of halogen
chemistry for MSA production (Fig. A3b). These results are comparable with observationally
constrained estimates from ATom-4 flight campaigns, where ∼ 30% - 40% DMS was oxidized to
HPMTF along their flight tracks compared to 27.7% for the full branch of HPMTF in the present
work, as well as with previous modeling studies showing 33% HPMTF formation as terminating
product (Veres et al., 2020; Fung et al., 2022). MSA is produced mostly by aqueous phase
oxidation of MSIA by $O_3$ and OH according to the mechanism used here and has high abundance
near the Southern Ocean and Antarctic belt as reported by previous studies (Chen et al., 2018;
Hoffmann et al., 2016; Fung et al., 2022). The global burden of MSA decreases dramatically, from
19 Gg S for 'Base' to 9.2 Gg S for simulation MOD. The higher rate of major loss process or lower
rate of production of MSA from the aqueous phase reactions could be responsible for this reduction
in global budget (Fung et al., 2022).
3.4 Impact on aerosol size distributions
Following the percent change in simulated surface layer $SO_2$ and $SO_4^{2-}$ for modified DMS
chemistry (Fig. 7), we further explore how this expanded DMS oxidation chemistry impacts
modeled aerosol size distributions. Figure 9 shows the global mean surface-layer percent change
in the normalized aerosol number concentration for modified chemistry relative to the BASE
simulation, with and without cloud and aerosol HPMTF loss processes. The aerosol number
concentration decreases for the sub-80 nm diameter size bins for both simulations, especially
during the DJF months when cloud and aerosol loss pathways of HPMTF are included (MOD
case), demonstrating the negative impact of these processes on simulated new particle formation.
Without these processes included (as in case MOD_noHetLossHPMTF), percent changes are
lower relative to simulation MOD but similar in terms of direction of changes. On the other hand,
HPMTF lost to clouds and aerosols increases the simulated number of particles with diameter
above 100 nm in the MOD simulation, consistent with the increase in sulfate mass concentrations
shown in Fig. 7 and suggesting that HPMTF heterogenous loss promotes simulated particle growth
to diameters larger than 80-100 nm. The greater abundance of particles larger than 100 nm also
acts as a condensation sink, further suppressing nucleation and growth at smaller size ranges.

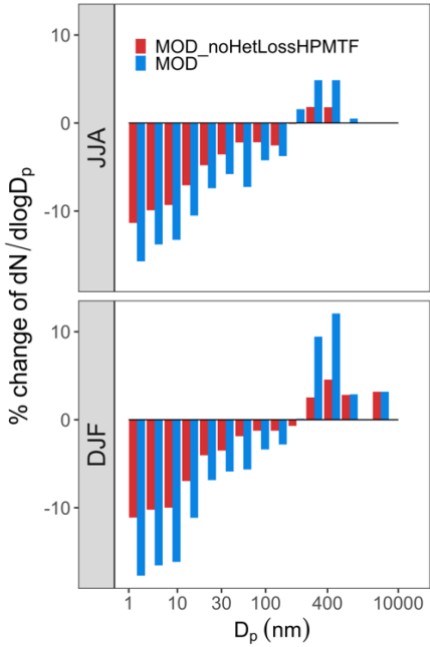

**Figure 9** Global mean surface-layer percent change in normalized aerosol number concentration for different size bins with particle diameter, $D_p$ in the range of 3 nm < $D_p$ < 10000 nm for simulations MOD and MOD_noHetLossHPMTF relative to simulation BASE. Simulations are described in Table 5.

The geographic distribution of surface layer aerosol number concentration for aerosol in the size
range of 3 – 80 nm for two seasons is shown in Figure 10. We find that global mean aerosol number
concentration in this size range decreases for simulations MOD and MOD_noHetLossHPMTF
relative to BASE by 16.8% and 11.7% respectively. Decreases are greater for simulation MOD
(Fig. 10b). Fig. 10c shows the effect of HPMTF heterogenous loss processes on the number of
particles with diameters between 3-80 nm for simulation MOD relative to simulation
MOD_noHetLossHPMTF. The largely negative impact of HPMTF loss to clouds and aerosols on
sub-80 nm particle number is contributed to by enhanced direct sulfate formation on pre-existing
particles, bypassing gas-phase $SO_2$ formation (a precursor for new particle formation). As well, in
the model, new particles grow through condensation of $H_2SO_4$ and organics and their growth are
dependent on the condensation sink, while loss of particle number depends on the coagulation
sink. Thus, changes to the condensation/coagulation sinks and sulfuric acid production rate
through the updated mechanism will also alter the growth rates of small particles (sub-80 nm) as
well as their coagulation loss rates. Hence, similar to the discussion for Figure 9, the reduction of
gas-phase production of $H_2SO_4$ in MOD relative to BASE slows new-particle formation and
growth, while the additional production of sulfate through aqueous chemistry on larger particles
in MOD increases the coagulational scavenging of the newly formed particles. These two effects
synergistically reduce the concentration of ultrafine particles in the model. The fraction of newly
formed particles that can reach the CCN size is dependent on the particle growth rates, especially
for particle sizes below 10 nm, where we see highest coagulation losses to larger particles. The
sensitivity of these results to the new sea salt debromination parameterization is shown in Fig. A4,
where we find a regional increase in aerosol number concentration at mid to higher latitude of the
SH despite low BrO concentrations (Fig A4).

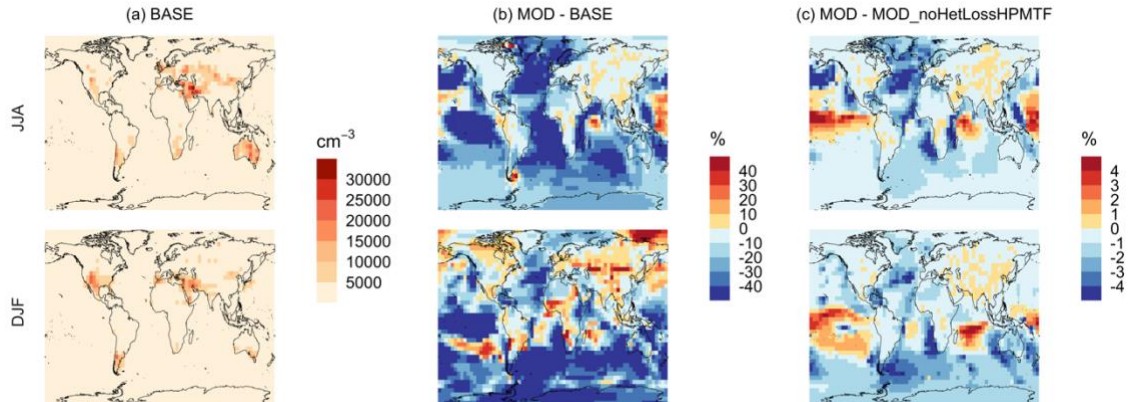

**Figure 10** Geographic distribution of seasonal-mean surface-layer aerosol number concentration in cm$^{-3}$ (for particles with diameters between 3 – 80 nm) for (a) the BASE simulation, (b) the percent difference between MOD and BASE and (c) the percent difference between MOD and MOD_noHetLossHPMTF to show the role of cloud and aerosol loss of HPMTF. The top and the bottom rows correspond to the months of JJA and DJF respectively. Simulations are described in Table 5.


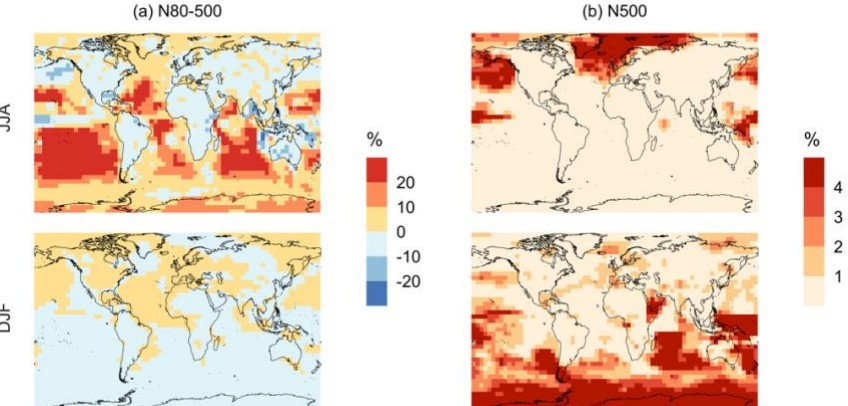

**Figure 11** Geographic distribution of percent difference in seasonal-mean surface-layer aerosol number concentration in cm$^{-3}$ for simulations MOD relative to simulations BASE for diameters between (a) 80 – 500 nm and (b) >500 nm. Simulations are described in Table 5.

Finally, we also analyze the impact of this expanded DMS scheme on particles larger than 80 nm
(Fig. 11). We find increases of around 6.7% for JJA mean surface layer number concentration of
aerosol with diameters between 80-500 nm, while DJF months show mean reductions of -5.4% for
DJF despite largely positive changes in the marine NH for these months (Fig. 11a). However, for
the > 500 nm size ranges (Fig. 11b), the global mean surface layer number concentration of aerosol
mostly increases, with highest changes occurring in the areas of peak DMS emission in both
hemispheres, during their summertime season. A similar trend is observed in the absence of cloud
and aerosol HPMTF uptake in simulation MOD_noHetLossHPMTF (Fig. A5). Overall, the global
annual mean number of particles with diameter larger than 80 nm increases by about 3.8%.
Comparing the regional extent and direction of change in particle number concentration, we find
the net increase in particle number concentration is higher for MOD compared to
MOD_noHetLossHPMTF, highlighting the importance of HPMTF loss processes to clouds and
aerosols as a contributor of CCN.

## 4    Conclusion

In this study we update the default DMS oxidation scheme in the GEOS-Chem model by implementing an integrated oxidation mechanism. The new scheme includes gas-phase and aqueous phase reactions involving DMSO, MSIA and HPMTF formation, as well as newly identified HPMTF loss processes yielding considerable changes in seasonal concentrations of major oxidation products and sulfur-derived aerosols. With this new chemistry scheme, global annual mean surface DMS concentration decreases by 36% relative to the BASE scheme in GEOS-Chem globally due to the presence of additional loss processes in the integrated mechanism reducing the bias to ATom-4 DMS measurement.

In this new scheme, OH, BrO, $O_3$ and $NO_x$ species act as important sinks of DMS contributing to 66.2%, 18.4%, 3.1% and 11.2% global annual mean surface DMS loss, highlighting the relative importance of these loss process in determining surface DMS budget. We also find that at higher latitudes, gas phase and multiphase oxidation of DMS by $O_3$ and BrO becomes important to determine the budget of DMS. On the other hand, overall OH is responsible for major loss of DMS via the addition and abstraction reaction relative to other sinks with more contribution from the addition reaction compared to abstraction reaction. For the global distribution of simulated HPMTF, our updated scheme in GEOS-Chem provides a reduced high bias against observations compared to previous studies. While emissions of BrO are uncertain in this version of GEOS-Chem, we find that the compound acts as a key sink of DMS, especially over the Southern Ocean. Overall, we find large reduction in $SO_2$ (40%) and an increase in sulfate (17%) due to the addition of heterogeneous HPMTF loss processes.

The lower $SO_2$ with the new DMS chemistry scheme contributes to a reduction in the global annual mean surface layer number concentration of particles with diameters less than 80 nm by 16.8%, contributed to by reductions in gas-phase precursors for new particle formation. There is a concurrent increase of 3.8% in the global annual mean number of particles with diameters larger than 80 nm. This latter global mean particle number change varies in sign seasonally, with a 6.7% increase for JJA, and a 5.4% decrease for DJF. This decrease is dominated by southern hemisphere summertime changes, connected with suppressed new particle formation/growth and enhanced coagulation following additional sulfate production through aqueous chemistry. Cloud loss processes related to HPMTF make key contributions to these simulated changes through enhancement of aqueous-phase particle growth of those particle large enough to act as CCN.

Although the increased chemical mechanism complexity described in this work will necessarily increase model computational cost (MOD simulation run times increase by approximately 16%), this study highlights the value of including a more realistic chemical oxidation mechanism of DMS and its stable intermediates for better representation of DMS-derived aerosol in the marine atmosphere, as well as its seasonal size distributions. A reduced form of the key chemical species and pathways should be able to capture the key processes with less computational impact and will be a priority in future work.

**Appendix A: Additional figures**

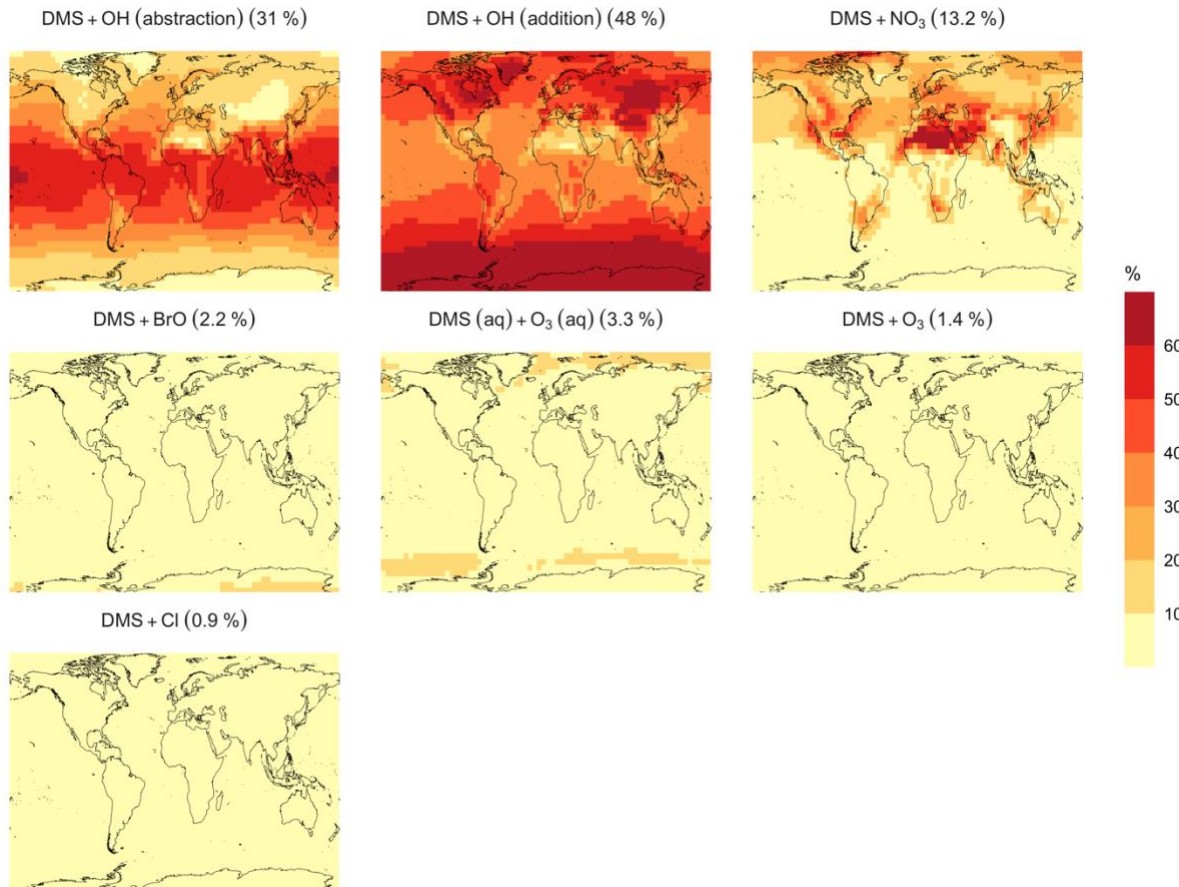

**Figure A1** Surface layer geographic distribution of the simulated annual mean fraction of total DMS oxidation (percent) attributed to different tropospheric oxidants for a simulation otherwise the same as simulation MOD except with no sea salt debromination. Percentages in parentheses indicates average contribution to global chemical loss as a fraction of DMS emitted for each reaction pathways presented here. Simulations are described in Table 5.


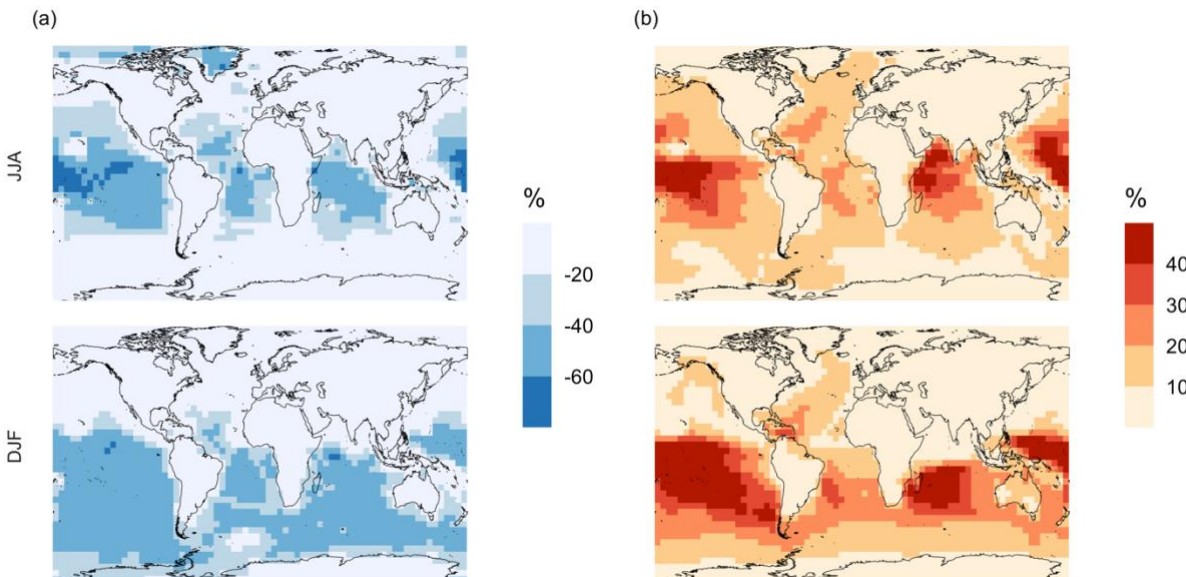

**Figure A2** Percent change in simulated surface layer (a) $SO_2$ and (b) $SO_4^{2-}$ for simulation MOD relative to MOD_noHetLossHPMTF for June, July and August mean (JJA) and December, January, and February mean (DJF). Simulations are described in Table 5.


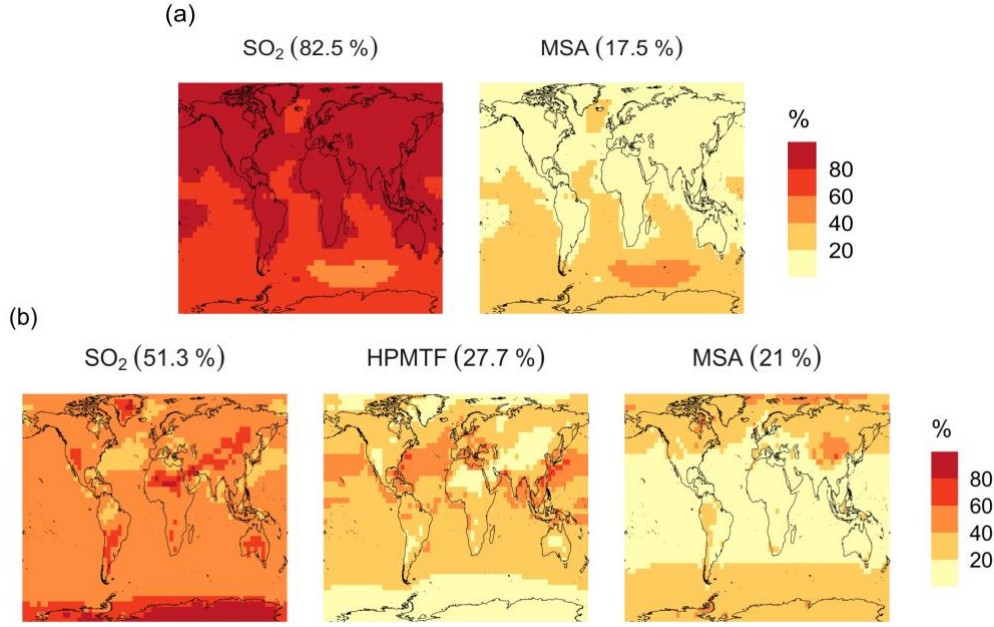

**Figure A3** Simulated annual mean surface layer branching ratios (in %) of the DMS oxidation mechanism considering $SO_2$, HPMTF, and MSA as major oxidation products calculated from their total production rates for simulations similar to (a, top row) BASE and (b, bottom row) MOD, except MOD with no sea salt debromination. Simulations are described in Table 5.


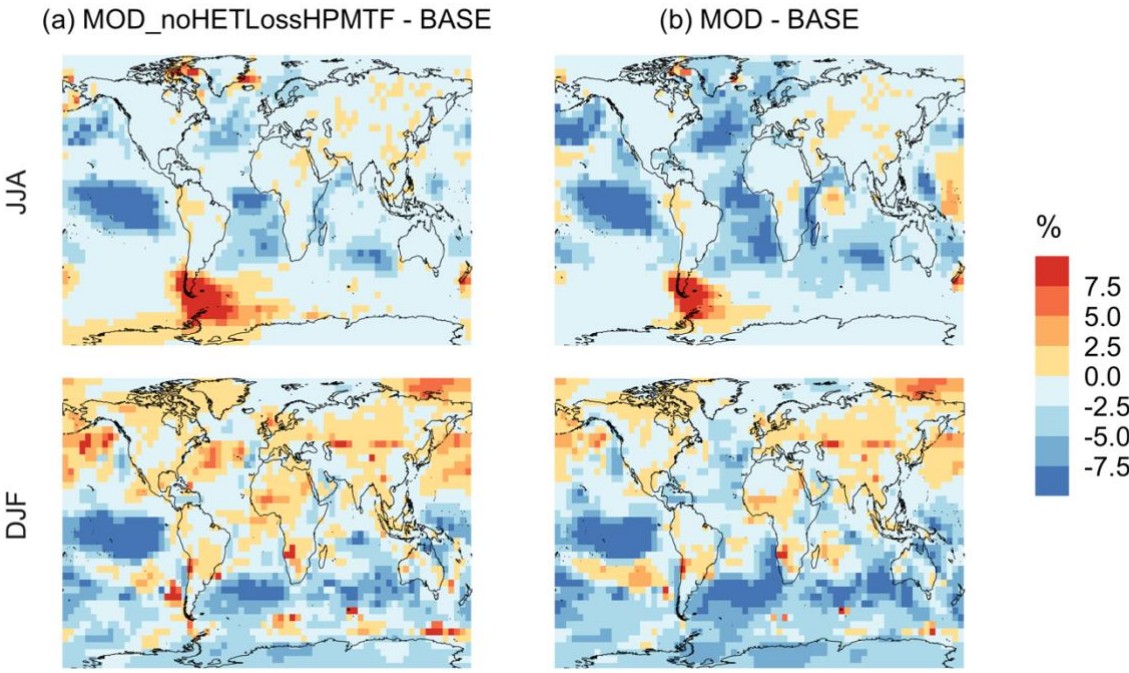

**Figure A4** Geographic distribution of percent difference in seasonal-mean surface-layer aerosol number concentration in cm-3 (for particles with diameters between 3 – 80 nm) for simulations similar to (a) MOD_noHetLossHPMTF and (b) MOD relative to simulations BASE, except all with no sea salt debromination. Simulations are described in Table 5.


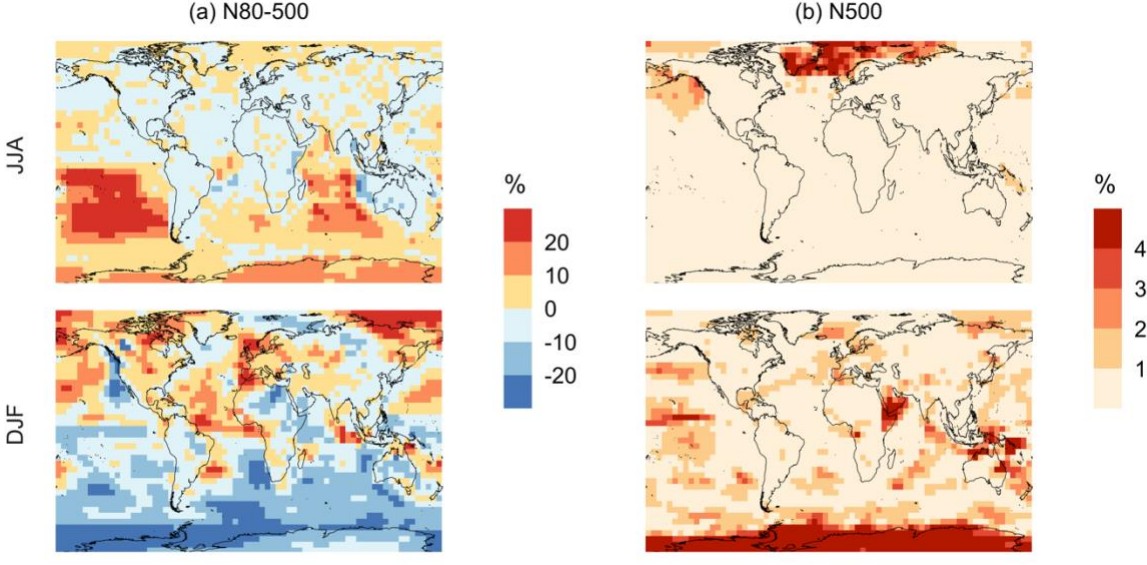

**Figure A5** Geographic distribution of percent difference in seasonal-mean surface-layer aerosol number concentration in cm$^{-3}$ for simulations similar to MOD_noHetLossHPMTF relative to simulations BASE, for particle diameters between (a) 80 – 500 nm and (b) > 500 nm. Simulations are described in Table 5.


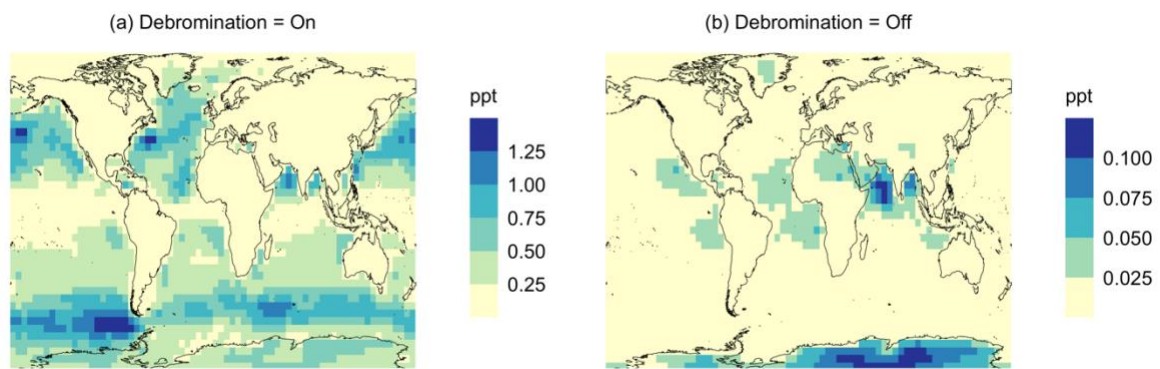

**Figure A6** Geographic distribution of mean surface BrO mixing ratio (ppt) for (a) with sea salt debromination and (b) without sea salt debromination for simulation MOD. Simulations are described in Table 5.


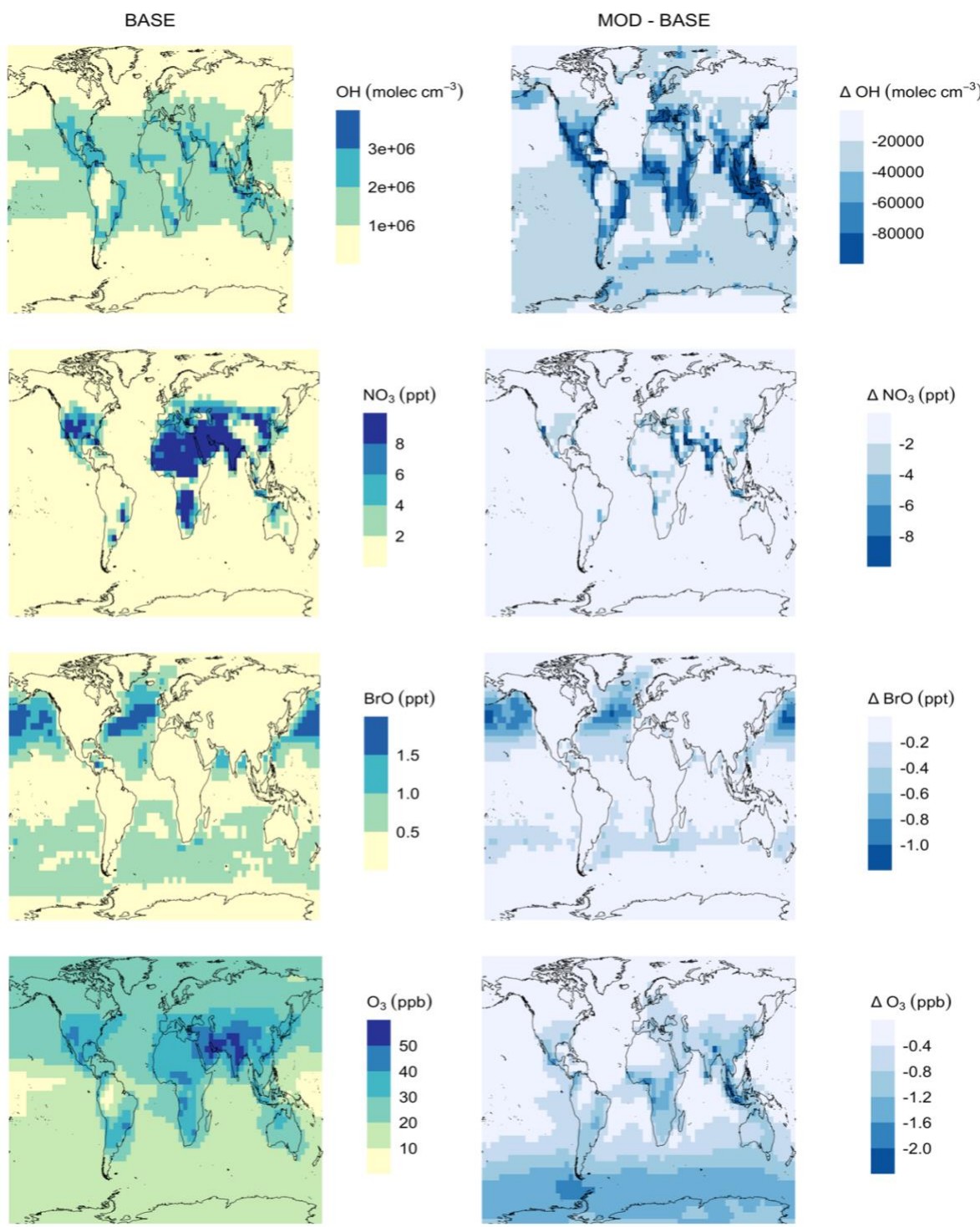

**Figure A7** Geographic distribution of mean surface oxidant concentrations for simulation (a) BASE and (b) MOD - BASE. Simulations are described in Table 5.


 **Appendix B: Additional Table**

**Table B1.** Global atmospheric flux, deposition, burdens, lifetime of DMS and its oxidation products, chemical loss rates for specific reaction pathways and global mean concentration of major oxidants are listed here for the case of simulation MOD. Note that $SO_2$ and $SO_4^{2-}$ includes natural as well as anthropogenic sources.

| | |
|---|---|
| $F_{DMS}$ (Gg S yr$^{-1}$) | $2.2 \times 10^4$ |
| Deposition of MSA (Gg S yr$^{-1}$) | $3.6 \times 10^3$ |
| Deposition of HPMTF (Gg S yr$^{-1}$) | $9.1 \times 10^1$ |
| Deposition of DMSO (Gg S yr$^{-1}$) | $1.7 \times 10^3$ |
| Deposition of MSIA (Gg S yr$^{-1}$) | $2.1 \times 10^2$ |
| DMS (GgS) | 65 |
| MSA (GgS) | 9.2 |
| HPMTF (GgS) | 0.6 |
| $SO_2$ (GgS) | 256.7 |
| $SO_4^{2-}$ (GgS) | 612.4 |
| $\tau_{DMS}$ (d) | 0.9 |
| $\tau_{MSA}$ (d) | 0.9 |
| $\tau_{HPMTF}$ (d) | 0.6 |
| $\tau_{SO2}$ (d) | 1.3 |
| $\tau_{SO4}^{2-}$ (d) | 4.4 |
| DMS lost to MSA (Gg S yr$^{-1}$) | $4.3 \times 10^3$ |
| DMS lost to HPMTF (Gg S yr$^{-1}$) | $6.9 \times 10^3$ |
| DMS lost to $SO_2$ (Gg S yr$^{-1}$) | $9.5 \times 10^3$ |
| MSA lost to particle growth (Gg S yr$^{-1}$) | $4.5 \times 10^2$ |
| HPMTF lost to $SO_2$ (Gg S yr$^{-1}$) | $4.8 \times 10^2$ |
| HPMTF lost to cloud (Gg S yr$^{-1}$) | $6.7 \times 10^3$ |
| HPMTF lost to particle growth (Gg S yr$^{-1}$) | $2.8 \times 10^2$ |
| OH (molec cm$^{-3}$) | $8.0 \times 10^5$ |
| $NO_3$ (ppt) | 0.97 |
| $O_3$ (ppb) | 21.10 |
| BrO (ppt) | 0.31 |

**Data availability.** The DMS observational data in Fig. 2 were obtained from the referenced papers (Kouvarakis and Mihalopoulos, 2002; Castebrunet et al., 2009). The observations data during ATom-4 are published through the Distributed Active Archive Center for Biogeochemical Dynamics (DAAC) at (Novak et al., 2021; Wollesen de Jonge et al., 2021), https://doi.org/10.3334/ORNLDAAC/1921 and https://daac.ornl.gov/ATOM/guides/ATom_SO2_LIF_Instrument_Data.html.

**Author contributions.** LT and WCP designed the research goals, aims, and methodology, implemented the new code into GC-TOMAS. QC, BA, CHF and CDH contributed in code development. All authors provided expert advice on data analysis, interpretation, and visualization. LT ran model simulations, analyzed the data, created the figures, and led manuscript development and editing.

**Competing interests.** The contact authors have declared that none of the authors has any competing interests.

**Acknowledgements.** LT and WCP gratefully acknowledge Ka Ming Fung for discussions on DMS oxidation chemistry. BC thanks Rachel Y.-W. Chang for discussions on marine aerosols.

**Financial support.** LT and WCP was supported by NSF grant no. 2155192. QC was supported by the Hong Kong Research Grants Council (Grant No. 15223221 and 15219722). BA was supported by NSF AGS 2109323 and PLR 1904128. CHF was supported by NASA FINESST (grant 80NSSC19K1368). CDH acknowledges funding support from NSF AGS (grant 1848372). BC gratefully acknowledges research funding supported by the Ocean Frontier Institute, through an award from the Canada First Research Excellence Fund. JRP was supported by the Atmospheric System Research (ASR) program, part of the US Department of Energy's Office of Biological and Environmental Research within the Office of Science, under grant DE-SC0021208. SI was supported by Ferring Pharmaceuticals through the Extreme Environments research Laboratory, École Polytechnique Fédérale de Lausanne (EPFL).

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
