# Peer review of "of DMS oxidation products and size-resolved sulfate aerosol"

_EGUsphere, 2023_

## Author Comment (AC1)

**Response to Referee 3 (RC3)**

Tashmim et al. report results from the GEOS-Chem global chemical transport model incorporating many recent findings on reactive intermediates in dimethyl sulfide (DMS) oxidation chemistry and quantify impacts on terminal products and aerosol particle size and abundance. This type of integrated analysis is necessary for evaluating the combined impact of the numerous recent revisions to our understanding of DMS chemistry and this work therefore has a high potential value. However, this is dependent on a thoughtful synthesis of reaction mechanisms from various sources which I believe needs some further work in this manuscript. In particular, I have concerns about how details of the reaction of DMS with $NO_3$ and Cl were implemented (see major comments below). Otherwise I find the work to generally be of a high quality and the results and discussions are well supported. If these apparent issue with the reaction mechanism are resolved along with the other comments below, then I believe this manuscript will likely be suitable for publication in *ACP*.

**Response:** We thank the Reviewer for positive feedbacks and helpful comments. Please find our point-by-point responses provided below.

**Major Comments:**

1.) My primary concern with the manuscript is what appears to me to be a mistake in the reaction mechanism resulting in the DMS + NO3 and DMS + Cl reactions being included twice, which impacts all of the results presented in this analysis. In table 2 the following reaction is listed:

DMS + NO3 → SO2 + HNO3 + CH3O2 + CH2O

rate: 1.90e-13*exp(530/T) reference: (Burkholder et al., 2015)

And in table 4 the following reaction is listed:

DMS + NO3 → MSP + HNO3

rate: 1.9e-13*exp(520/T) references: (Novak et al., 2021; Wollesen de Jonge et al., 2021)

These are not two distinct chemical reactions. Both reactions are an H-abstraction from DMS by NO3 with the same rate constant. The only difference is in the assigned products where the reaction in table 2 makes the simplifying assumption that SO2 is formed at unit yield, while table 4 instead goes through the reactive intermediate species MSP. In reality the reaction in table 2 also proceeds through MSP, this was likely just neglected in the referenced compilation of Burkholder et al., 2015 because the significance of the MSP intermediate for HPMTF chemistry was not know at the time of that data evaluation. Only the Reaction in table 4 should be included in the model. By including both you are double counting this reaction pathway and incorrectly increasing the modelled significance of NO3 chemistry.

**Response:** We appreciate the catch and have removed the reaction DMS + $NO_3$ → $SO_2$ + $HNO_3$ + $CH_3O_2$ + $CH_2O$ (Table 2), keeping only the reaction DMS + $NO_3$ → $CH_3SCH_2OO$ (MSP) +

HNO₃ (Table 4) in our revised chemical mechanism, following MCMv3.3.1. and other recent modeling studies (e.g. Novak et al., 2021, Wollesen de Jonge et al., 2021, Novak et al., 2022).

Similarly, for DMS + Cl the following reaction is given in Table 2:

DMS + Cl → 0.5SO2 + 0.5DMSO + 0.5HCl + 0.5ClO

rate: 3.40e-10 reference: (Barnes et al., 2006; Burkholder et al., 2015)

While in Table 4 the following reaction is listed:

DMS + Cl → 0.45MSP + 0.55C2H6SCl + 0.45HCl

Rate: 3.40e-10 reference: (Fung et al., 2022)

Again, these are fundamentally the same reaction resulting in this chemistry being double counted. The only difference is in the simplifying assumptions made about product yields.

**Response:** We removed DMS + Cl —> 0.5SO₂ + 0.5DMSO and kept DMS + Cl = 0.45 MSP + 0.55C₂H₆SCl + HCl to keep the products from addition and abstraction channel. We also updated the rate constant to 3.6e-10 for this reaction according to IUPAC recommendation. In addition to that, we have changed the C₂H₆SCl chemistry. Instead of having a null cycle where C₂H₆SCl decompose to DMS via the reaction C₂H₆SCl → DMS + Cl, it now continues the oxidation as, C₂H₆SCl = DMSO + ClO. With these reactions we find very small change in our results for this loss process shown in Figure 6 and Figure A1 for DMS + Cl channel.

2.) The results of Jernigan et al. (2022) show that HPMTF is the primary precursor to OCS formation from DMS oxidation with HPMTF + OH -> 0.13 OCS + 0.87 SO2. At a minimum, this should be considered as it will reduce the overall SO₂ production from DMS oxidation which will impact the results presented in this manuscript. The overall yield of OCS is also therefore highly dependent on HPMTF multiphase loss processes. With minimal additional analysis, this work could also provide a valuable update on to the GEOS-Chem modeling results from Jernigan et al. (2022). I do not feel strongly that extended analysis of OCS production should be included, but do feel that some comment on the impacts on SO2 production are necessary beyond what is included at lines 91-95.

**Response:** Previously we had an indirect OCS formation pathway in the model as follows:

C₂H₃O₃S = OH + CH₂O + OCS (see Table 3)

Followed by Jernigan et al., 2022a, in our revised mechanism, beside this reaction mentioned earlier, we added the following reactions as OCS formation and loss pathway as part of the DMS oxidation mechanism.

    a) HPMTF + OH → 0.13 OCS + 0.87 SO₂ + CO
    b) OCS + OH → SO₂

With these two additional reactions, we find that rather than 38% reduction in $SO_2$ formation now we have 35% reduction in $SO_2$ due to stepwise conversion of HPMTF to $SO_2$ via OCS, although yield of $SO_2$ from non-HPMTF pathway decreases to 45.3% from 52.4%. Note that we have made changes to few other reactions in the revised mechanism which does not involve OCS, so these changes in percentage of $SO_2$ might not be entirely attributed to the reactions involving OCS. We have added a description of this near lines 102-106:

"With the latest experimental findings on heterogeneous loss process of HPMTF and experimentally validated oxidation reactions for OCS formation directly from HPMTF it is necessary to include these reactions as part of the DMS oxidation mechanism as these will have impact on overall yield of $SO_2$, thus affecting the formation probability of CCN (Jernigan et al., 2022a, b)."

Lines 471-475:

"One of the reactions that possibly contributes to delayed formation and reduction of $SO_2$ concentration is the first-generation OCS formation from OH oxidation of HPMTF. We find that addition of cloud and aerosol loss significantly decreases the OCS production, especially at the high cloud cover region as previously reported (Jernigan et al., 2022a)"

3.) $SO_2$ mixing ratios were measured during the ATom-4 campaign at suitable precision to be informative in background marine air masses (https://daac.ornl.gov/ATOM/guides/ATom_SO2_LIF_Instrument_Data.html). A comparison of measured and modelled SO2 could be a very useful addition.

**Response:** We have added this comparison with explanation in Figure 2c. Line 340-355:

"We also compared the $SO_2$ concentrations measured during ATom-4 by Laser Induced Fluorescence (LIF) and simulation MOD values for nearest neighbor grid cells are shown in Figure 3c across different altitude. Modeled surface $SO_2$ concentrations are lower than those observed during ATom-4 missions across the vertical scale shown here for simulation MOD. The greater $SO_2$ losses results in a shorter $SO_2$ lifetime (from 1.4 d to 1.3 d) for simulation MOD relative to simulation BASE. The reduction in modeled $SO_2$ is largest over the Southern Ocean (shown later in Fig. 7a) where heterogeneous oxidation of HPMTF is most efficient and irreversible. Removing the heterogeneous loss of HPMTF increases the modeled $SO_2$ compared to simulation MOD with underprediction remaining for altitudes > 1km. Remaining model biases could be at least partially attributed to uncertainty in DMS oxidation processes along with other non-DMS sources contributing high concentration of $SO_2$. Aside from uncertainty in DMS emissions and oxidation, recent understanding of marine sulfur chemistry such as methanethiol ($CH_3SH$) oxidation has been reported as an significant source of $SO_2$ in the marine atmosphere and could help reduce the bias, a possibility deserving further investigation (Berndt et al., 2023; Novak et al., 2022). Overall the DMS oxidation chemistry implemented in this work reduces the model observation bias close to the surface (up to 1km) compared to BASE GEOS-Chem chemistry."

**Other Comments:**

What SO₂ heterogenous chemistry is included in this work?

**Response:** We do have cloud chemistry of $SO_2$ in the default version of GEOS-Chem (v12.9.3) which includes in-cloud oxidation of $SO_2$ by $H_2O_2$, $O_3$ and $O_2$ catalyzed by transition metals (Mn, Fe) as well as oxidation by HOBr and HOCl. Since this is not a new addition to the model, we did not highlight that in our manuscript. To address this comment, we have added line 258-263 in the revised version as:

"Alongside the gas-phase and aqueous-phase reactions relevant to the added DMS oxidation mechanism contributing to the formation of $SO_2$ and sulfate, the default version of GC-TOMAS used here also includes in-cloud oxidation of $SO_2$ by $H_2O_2$, $O_3$, and $O_2$ catalyzed by transition metals (Mn, Fe), as well as the loss of dissolved $SO_2$ by HOBr and HOCl, all of which are passed to TOMAS to account for sulfate production (Chen et al., 2017; Wang et al., 2021)."

The sensitivity runs with and without sea-salt aerosol debromination are appreciated given remaining uncertainties in BrO measurements and model implementations. Is it correct that the revised debromination mechanism of Wang et al. 2021 was not used here? If so what is the motivation for this? This comment is based on the references included in the methods section in lines 171-172.

**Response:** We have used the default debromination mechanism available for GEOS-Chem v12.9.3 and that does not include revisions from Wang et al. 2021. We did included sensitivity runs with and without sea-salt aerosol debromination just to evaluate the contribution of this process in resolving some uncertainty associated with BrO.

Can you show a figure of the global distribution of BrO in the MOD and MOD without sea salt debromination model cases? Otherwise it is difficult for the reader to make absolute comparisons for either model case to measurements of BrO.

**Response:** We have added a global distribution of BrO for MOD and MOD without sea salt debromination as Figure A6 and mentioned that in lines 428 – 431:

"As would be expected, these simulations show much lower BrO formation (as shown in Fig. A6) and resulting chemical impacts, with overall oxidation contributions comparable to previous literature (Schmidt et al., 2016; Wang et al., 2021)."

Table 3 and lines 224 - 230: Jernigan et al. (2022) provides an experimental value for k(HPMTF + OH) of 1.4E-11 cm3 molec^-1 s^-1 which is a useful validation of the assumed value of 1.1E-11 cm3 molec^-1 s^-1 used here and in Vermeuel et al. (2020) and Novak et al. (2021). This should be referenced.

**Response:** In the revised manuscript we used the experimentally determined rate constant of $1.40 \times 10^{-11}$ $cm^3$ $molecules^{-1}s^{-1}$ for this reaction and addressed this in Table 3 and near line 237-240 along with proper reference suggested here. The change to the manuscript involves line 237-240:

"We use a rate constant of $1.40 \times 10^{-11}$ cm$^3$ molecules$^{-1}$s$^{-1}$ for HPMTF + OH, which was previously determined based on concentrations of other known sulfur species (DMS, DMSO, SO$_2$ and methyl thioformate; MTF; CH$_3$SCHO; a structurally similar proxy to HPMTF) and evaluated by box model (Jernigan et al., 2022a)."

Line 388 and onward: You should make clear what the altitude range is for the quoted reductions and enhancements are in the simulation. Presumably these are for some near surface range and not total column?

**Response:** Correct, this is not for the total column. We have added the term 'surface layer' at line 453 and elsewhere while mentioning such numbers which represents the vertical level 1 of GEOS-Chem.

Figure 9. It appears that much of the particle number increase is for Dp > 200 nm. What is the size range where CCN abundance is most sensitive to particle growth? Some additional context for the reader may be useful in connecting changes in particle size bins to potential changes in CCN abundance.

**Response:** For better understanding we added line 536-538:

"The fraction of newly formed particles that can reach the CCN size is dependent on the particle growth rates, especially for particle sizes below 10 nm, where we see highest coagulation losses to larger particles."

---

## Author Comment (AC2)

**Response to Referee 1 (RC1)**

Tashmim et al report global model simulations of DMS oxidation. The model includes an advanced DMS oxidation scheme that accounts for recent insights into DMS oxidation chemistry. This work builds on the work of Novak et al., where the gas and multiphase chemistry of HPMTF was explored initially in GEOS-Chem. This work significantly advances beyond the study of Novak et al. to investigate the role of $DMS+O_3$ multiphase chemistry and the impact of the new DMS oxidation mechanisms on particle number and size distributions. The manuscript is well written and should be published following the authors attention to the following points:

**Response:** We thank the reviewer's positive and constructive feedback. We have addressed and revised the manuscript accordingly. Our point-by-point responses are provided below.

**General comments:**

The modified version of the model has HPMTF and DMS cloud chemistry. It was not clear to me how DMSO and $SO_2$ cloud chemistry was treated in the model. This seems to be an important component of sulfur cycling in the MBL that could be addressed here. Was this included, but not discussed or was this chemistry not included in the revised mechanism.

**Response:** Beyond the HPMTF and DMS cloud chemistry additions we have made, the default version of GEOS-Chem (v12.9.3) does include in-cloud oxidation of $SO_2$ by $H_2O_2$, $O_3$, and $O_2$ catalyzed by transition metals (Mn, Fe). Since this is not a new addition to the model, we did not highlight that in our manuscript earlier. To clarify the treatment of DMSO, for the reactions of $DMS(g) + O_3(aq)$, $DMSO(g) + OH(aq)$, $MSIA(g) + OH(aq)$, $MSIA(g) + O_3(aq)$ and $MSA(g) + OH(aq)$ in cloud droplets and aerosols, we assume a first-order loss of the gas-phase sulfur species following the parameterization described in Chen et al. 2018 and original references, and we use the same values for physical parameters that were used in that study. To address this comment and better explain these details, we have added the following text in the revised manuscript in lines 255-263:

"For the aqueous-phase reactions listed in Table 2, including the oxidation of intermediates DMSO and MSIA in cloud droplets and aerosols, a first-order loss of the gas-phase sulfur species was assumed following previously used parameterizations and physical parameter values (Chen et al., 2018). Alongside the gas-phase and aqueous-phase reactions relevant to the added DMS oxidation mechanism contributing to the formation of $SO_2$ and sulfate, the default version of GC-TOMAS used here also includes in-cloud oxidation of $SO_2$ by $H_2O_2$, $O_3$, and $O_2$ catalyzed by transition metals (Mn, Fe), as well as the loss of dissolved $SO_2$ by HOBr and HOCl, all of which are passed to TOMAS to account for sulfate production (Chen et al., 2017; Wang et al., 2021)."

The percentage of DMS lost to each reaction pathway (e.g., OH, BrO, $O_3$(aq), $NO_3$) is cited in the conclusions and features in figure. It is not abundantly clear how these percentages were calculated. Are these the fraction of DMS emitted that is lost to each of these reaction pathways? Or is this the average of the fractional losses (e.g., f(DMS_OH)/total loss) averaged spatially over the entire map? I think it should be (and probably is) the former, but it would be helpful to have confirmation.

**Response:** This percentage indicates the former case. To avoid confusion, we have updated the captions for Figure 6 and Figure A1 to clarify that the percentage here indicates the fraction of DMS lost to each of this specific reaction pathways.

"**Figure 6** Geographic distribution of the annual mean surface layer fraction of total DMS oxidation (percent) attributed to different tropospheric oxidants for simulation MOD (described in Table 5). Percentages in parentheses indicates average contribution to global chemical loss as a fraction of DMS emitted for each reaction pathways presented here."

"**Figure A1** Surface layer geographic distribution of the simulated annual mean fraction of total DMS oxidation (percent) attributed to different tropospheric oxidants for a simulation otherwise the same as simulation MOD except with no sea salt debromination. Percentages in parentheses indicates average contribution to global chemical loss as a fraction of DMS emitted for each reaction pathways presented here. Simulations are described in Table 5."

**Specific Comments**

Line 48: Cl and BrO should be in parentheses rather than brackets.

**Response:** We have replaced the brackets with parentheses in line 49 for Cl and BrO.

Line 81: Are you referring to a multiphase DMS+OH addition pathway or multiphase DMSO chemistry, or a DMS+O3 pathway. In either case, it would be helpful to be specific.

**Response:** We were referring to OH-addition pathway of DMS forming DMSO and MSIA as the intermediates. We rephrased the lines 83-87 as:

"For example, the OH-addition pathway of DMS forms DMSO and MSIA as the intermediates, which has been identified as a dominant source of MSA via their aqueous-phase oxidation, and a fraction of that MSA subsequently undergoes aqueous-phase oxidation to form sulfate aerosol (Chen et al., 2018; Ishino et al., 2021; Zhu et al., 2006; von Glasow and Crutzen, 2004)."

Line 87: I appreciate that MSP is used in the literature, but I don't know why. What is that an acronym for? I would suggest MTMP.

**Response:** We have followed the nomenclature of Fung et al. 2022, which used MSP in reference to the methylthiomethyl peroxy radical ($CH_3SCH_2OO\cdot$). We do note that Jernigan et al., 2022a did use the abbreviation MTMP for the same species. Considering that this is a relatively recently identified species, and to maximize clarity in the absence of a fixed standard, we mention both of these alternatives initially and then continue with the more commonly used (so far) abbreviation of MSP throughout the rest of the manuscript. This is written in lines 89-94:

"More recent experimental and laboratory studies have confirmed the formation of methylthiomethyl peroxy radicals ($CH_3CH_2OO$; abbreviated MSP or MTMP) from the H-abstraction channel of OH oxidation, which can subsequently lead to a series of rapid intramolecular H-shift isomerization reactions, ultimately resulting in the formation of the stable intermediate HPMTF (Berndt et al., 2019; Veres et al., 2020; Vermeuel et al., 2020; Wu et al., 2015; Fung et al., 2022; Jernigan et al., 2022a)."

Line 90: I don't think all of these references are for the last statement (30-50% of DMS ends up as HPMTF). Perhaps distribute the references through the sentence so they refer to the correct statements?

**Response:** We have rephrased this line and broken it down into two sentences to properly indicate associated references. This is now written in lines 89-98:

"More recent experimental and laboratory studies have confirmed the formation of methylthiomethyl peroxy radicals ($CH_3CH_2OO$; abbreviated as MSP or MTMP) from the H-abstraction channel of OH oxidation, which can subsequently lead to a series of rapid intramolecular H-shift isomerization reactions, ultimately resulting in the formation of the stable intermediate HPMTF (Berndt et al., 2019; Veres et al., 2020; Vermeuel et al., 2020; Wu et al., 2015; Fung et al., 2022; Jernigan et al., 2022a). It has been reported that 30–46% of emitted DMS forms HPMTF according to different modeling studies and this falls within the observational range from NASA Atmospheric Tomography ATom-3 and ATom-4 flight campaigns where about 30–40% DMS was oxidized to HPMTF along their flight tracks (Fung et al., 2022; Veres et al., 2020; Novak et al., 2021)."

Figure 1: In my version there are no green boxes as referenced in the figure caption (they are orange). Are these the only species and reactions used? More specifically, is DMSO chemistry included? It is discussed in the text surrounding Figure 1, but not highlighted in the figure caption. I appreciate that this may complicate the figure (and I am not suggesting it needs to be added), but if DMSO features in the model, it would be good to state it in the figure caption.

**Response:** Thank you for spotting this mismatch between the figure and its caption. We have updated the figure and caption using only blue boxes for the three major oxidation products MSA, HPMTF and $SO_2$, all of which eventually oxidize to sulfate. We do have DMSO and MSIA in the model, which eventually contribute to $SO_2$ and MSA, but in the figure our intent was to highlight only the major oxidation products for simplicity of the scheme. However, to address this concern we have updated the Figure 1 caption to read:

"**Figure 1** Modified DMS oxidation mechanism used in this work (simulation MOD) showing the formation of major stable oxidation products (blue-outline boxes) including newly identified intermediate HPMTF, and their contribution to new particle formation or growth of existing particles. Note that we include gas-phase and aqueous-phase chemistry of DMSO, MSIA and OCS in the mechanism, which counts towards their contribution to the formation of major oxidation products of DMS, but that these pathways are not explicitly shown here to maintain visual clarity."

Line 125: The numbers cited here are from the global model simulation across all cloud fields, not just for the cloudy case. Perhaps this was the intent of the sentence, but maybe breaking this into two sentences would help get this point across that the 24% reduction in MBL $SO_2$ is a global, annual average not from the case study.

**Response:** We break this line into two sentences to better explain the meaning of these percentages. This is now rephrased in lines 133-138:

"Other work has used direct airborne eddy covariance flux measurements to explain the chemical fate of HPMTF in the MBL, finding that in cloudy conditions chemical loss due to aqueous phase reactions in clouds is the major HPMTF removal process (Novak et al., 2021). In the same study, global model simulations showed a 35% reduction in global annual average $SO_2$ production from DMS and a 24% reduction in the near-surface (0 to 3 km) global annual average $SO_2$ concentrations over the ocean as a result of this process (Novak et al., 2021)."

Table 4: What is MSP + MO2?

**Response:** In the earlier version MSP + MO2 means $CH_3SCH_2OO\cdot$ + $CH_3O_2$. Here, $CH_3O_2$. (Methylperoxy radical) is abbreviated as MO2 following the GEOS-Chem chemical mechanism. In the revised version we have replaced MSP + $MO_2$ with MSP + $CH_3O_2$ in Table 4 for clarity.

Table 5 caption: It would be helpful to fully explain what HPMTF =SO42- means. I think you mean there is a 100% S-yield of SO42-. Also, is gamma here really the activity coefficient? I think you mean uptake coefficient.

**Response:** By HPMTF = $SO_4^{2-}$ we mean loss of HPMTF via cloud and aerosol results in instant formation of $SO_4^{2-}$. However, losses in clouds follow entrainment-limited uptake which controls the rates of mixing between cloudy and clear air in the chemical rate expression (Novak et al., 2021; Holmes et al., 2019). Here, gamma ($\gamma$) is the reactive uptake coefficient for these loss processes. In the revised version we have used an experimental value of $\gamma$ for the aerosol loss of HPMTF and modified and corrected Table 5 footnote at lines 223-224 as:

"* Instantaneous formation of sulfate via HPMTF cloud and aerosol loss reactive uptake co-efficient ($\gamma$) of 0.0016."

Line 224: OH+HPMTF was measured in Jernigan et al. it would be best to cite that.

**Response:** In the revised manuscript we used the experimentally determined rate constant of $1.40 \times 10^{-11}$ cm$^3$ molecules$^{-1}$s$^{-1}$ for this reaction and corrected this near line 250-252 along with proper reference suggested here. The new line is line 237-240:

"We use a rate constant of $1.40 \times 10^{-11}$ cm$^3$ molecules$^{-1}$s$^{-1}$ for HPMTF + OH, which is determined based on concentration of other known sulfur species (DMS, DMSO, SO$_2$ and methyl thioformate; MTF; CH$_3$SCHO; a structurally similar proxy to HPMTF) and evaluated using a box model (Jernigan et al., 2022a)."

Line 236: I don't think it maters at all (since loss is diffusion limited in the cloud) but the HPMTF uptake coefficient to dilute cloud droplets should not be faster than that to the aerosol. I would use the experimentally determined value from Jernigan for both. Again, I don't think it matters for the simulation.

**Response:** We have used reactive uptake coefficients ($\gamma$) of 0.0016 for both cloud and aerosol loss of HPMTF, which represents the experimentally determined value of $\gamma$(HPMTF) to deliquesced NaCl (Jernigan et al., 2022b). As predicted by the reviewer, we find little difference in percent of HPMTF lost to cloud since the loss is diffusion limited. However, on a fractional basis the percentage of HPMTF lost to aerosol does increase from 2.3% to 3.4%.

Line 280: How are these fractions of DMS loss calculated? Is this taking the map (in Figure 3) and calculating and average % or is this weighted by the amount of DMS that is lost. Given the strong spatial gradients in DMS I think this makes a difference.

**Response:** For line 407, the numbers mentioned as "full conversion yield of DMS into $SO_2$ (82.5%) and MSA (17.5%)" is presented in Fig. A3a. We have added the reference of this figure at the end of this line (lines 406-408 in the revised manuscript). This is for the case of BASE simulation and calculated by the fraction of DMS emitted that is lost as MSA and $SO_2$. On the other hand, Fig. 6 represents fraction of DMS emitted lost to each of the specific reaction pathways mentioned as the header of individual maps.

Line 286: What are the "two possible pathways" Shouldn't DMS+NO3 make MTMP with 100% yield? I am really surprised that DMS+NO3 accounts for 15% of the total DMS loss? That seems big to me as I'd expect [NO3] to be almost zero at the surface over the ocean. Perhaps some more discussion on this point is needed.

**Response:** We have removed the reaction $DMS + NO_3 \rightarrow SO_2 + HNO_3 + CH_3O_2 + CH_2O$ and kept only one $DMS + NO_3$ reaction which gives MSP with 100% yield. With that being the only loss process of DMS via $NO_3$, we find this reaction accounts for 12.8% of the total DMS loss with major loss happening in the NH coastal regions due to high $NO_x$ emission from nearby land-based sources. However, over the ocean this is mostly less than 10% except for upper to mid-latitude Northern Hemisphere. Note that previous modeling studies have reported even higher values for the global average percent loss of DMS by $NO_3$ (16% and 22.5% by Chen et al. 2018 and Fung et al. 2022). With the change in chemistry and associated results we have revised the main text to address this question near lines 412-415:

"$NO_3$ oxidation of DMS accounts for another 12.8% of global DMS chemical losses, comparable to values found in previous studies (Chen et al., 2018; Fung et al., 2022). Over the ocean the $NO_3$ loss pathway is strongest in the NH coastal regions due to outflow of $NO_x$ sources from over the land, whereas for the SH values are generally less than 10%."

Figure 5, Line 352: These DMS measurements look very, very low. I think it is appropriate to question whether they are correct. Also, what measurements are used to create Figure 5?

**Response:** Thank you for noticing this issue. We did find an error in processing the input data for ATom-4 comparison using the planeflight diagnostic of the model and fixed it in the revised manuscript. This error does not impact any results other than Figure 3, and resolving this issue has improved our comparisons with observations and other simulations. We have added two more model simulations output for this vertical profile in the revised manuscript which are BASE and MOD_noHetLossHPMTF. In Figure 3a of the revised manuscript, the DMS measurements shown are now comparable to other literature sources that have used the same measurements for model/observation comparisons (Fung et al., 2022; Novak et al., 2021). For Figure 5 (now Figure 3), we do mention in the main text that the measurements used are from ATom-4 aircraft observations on the NASA DC-8 aircraft. The measurements used here were done by Iodide CIMS, Whole Air Sampler (WAS) and Laser Induced Fluorescence (LIF) for HPMTF, DMS and $SO_2$ respectively, and the links to those datasets were provided under the 'Data Availability' section. We also revised the caption for Figure 3 as:

"**Figure 3** Vertical profiles of (a) DMS, (b) HPMTF and (c) $SO_2$ mixing ratios from ATom-4 observations (black) and model with simulation MOD sampled along the ATom-4 flight tracks (red) binned every 500 m of flight altitude. Also shown are modeled results without HPMTF heterogeneous loss with simulation MOD_noHetLossHPMTF (yellow), and for BASE GEOS-Chem chemistry (blue). Box plot whiskers show full range of distribution at each altitude bin. DMS observations are from Whole Air Samples (WAS) while HPMTF DC-8 observations are from iodide ion chemical ionization time-of-flight mass spectrometer (CIMS). $SO_2$ observations from ATom-4 campaign were measured by Laser Induced Fluorescence (LIF)."

Figure 6: Without constraining the DMS flux, I don't think it is possible to attribute the improvement in model-measurement of [DMS] to inclusion of DMS+BrO. It is very likely that the DMS emissions are driving this.

**Response:** We agree that DMS emissions play a crucial role in our comparison, as they vary considerably with changes in sea surface DMS climatology, and we acknowledge that improved and validated high-resolution inventories will be necessary to address some of these questions. Here we simply intend to highlight and explain changes between standard and modified chemistry, and to note that the impact of the DMS + BrO reaction is one possible contribution to improved model-measurement agreement. Fig. 5b shows that modeled losses of DMS are especially strong in the upper latitudes of both hemispheres, where DMS + BrO is shown to be an important chemical loss process. Thus, with identical (if imperfect) DMS emissions driving both BASE and MOD cases we can say that within the expanded mechanism DMS + BrO appears to play a meaningful role, reducing DMS concentrations compared to BASE and bringing them closer to observations. To better describe these results, in the revised manuscript we have rephrased and added lines 289-293:

"Similarly, for Amsterdam Island major overpredictions are apparent for the BASE simulation compared to MOD for the months of May-August. One reaction that may play a role in this shift

is DMS + BrO, which as indicated earlier is responsible for a faster overall chemical loss of DMS, in particular over the southern hemisphere high latitudes."

---

## Author Comment (AC3)

**Response to Referee 2 (RC2)**

This paper explores, within the GEOS-Chem CTM, the impact of a more complex description of the oxidation of DMS on the concentration of sulfur compounds and size resolved aerosol. This is an important area of research with the oxidation of DMS providing a significant, natural background source of sulfur in both the present and past atmospheres. Having a robust understanding of this chemistry is thus vitally important for us to understand both the present day atmospheres and any changes from the preindustrial to the present day. The current representation of the chemistry scheme in this model (the three reactions given in Table 1) is outdated and it is good to see that some development work is taking place.

I however I have two significant concerns about this paper and then a number of smaller ones (described below). Until these major concerns are addressed I don't think the paper is suitable for publication.

**Response:** We thank the reviewer's constructive feedback and detailed attention to the chemical mechanism we used here. We have revised the mechanism accordingly and addressed all the major and minor issues mentioned here. Our point-by-point responses are provided below.

**Major issues:**

- **The new chemistry scheme**. The mechanism used for the model is a merging of a number of different mechanisms available in the literature. However, I have some concerns about how this has been done.

The DMS + NO3 reaction appears to be in twice. It is in the OH addition pathway section and in the H-abstraction pathway section. The rate constant is the same for both pathways but is given different references. I think this essentially means that this reaction is double counted and the DMS+NO3 channel is twice as fast as it should be. Both the latest IUPAC and NASA data evaluation has a single NO3 + CH3SCH3 → CH3SCH2 + HNO3 reaction for this. Thus any subsequent chemistry needs to come from the further oxidation of CH3SCH2. Thus I think that there has been double counting by having this reaction in twice.

**Response:** We appreciate the catch and have removed the reaction DMS + NO$_3$ → SO$_2$ + HNO$_3$ + CH$_3$O$_2$ + CH$_2$O (Table 2), keeping only the reaction DMS + NO$_3$ → CH$_3$SCH$_2$OO (MSP) + HNO$_3$ (Table 4) in our revised chemical mechanism, following MCMv3.3.1. and other recent modeling studies (Fung et al., 2022; Novak et al., 2021).

Similarily, I am confused by the DMS+Cl reaction. It has two channels, an abstraction channel (DMS+Cl◊CH3SCH2 + HCl) and an addition channel (DMS+Cl◊DMS-Cl). The IUPAC recommendation gives the recommendation of 3.6e-10 for both reactions with a 50:50 ratio between the two. This paper seems to follow this recommendation with a reaction of DMS+Cl◊0.5SO2+0.5DMSO+0.5HCl+0.5ClO. However, an additional reaction DMS+Cl◊0.45MSP+0.55C2H6SCl+0.45HCl is also included in the scheme. This is again is a split between the addition and abstraction reactions (0.55:0.45). But it appears that the overall DMS+Cl reaction is in the mechanism twice. I'm also then a bit confused by the C2H6SCl chemistry. I think the only thing that can happen to this in the mechanism is that it falls apart back to DMS+Cl. Thus

the addition channel in this part of the chemistry is effectively a null cycle for DMS oxidation whereas for the other DMS+Cl reaction there is an assumption that it leads to the continued oxidation of the DMS.

**Response:** We have removed DMS + Cl —> 0.5SO$_2$ + 0.5DMSO and kept DMS + Cl = 0.45 MTMP + 0.55C$_2$H$_6$SCl + HCl to keep the products from the addition and abstraction channels. We have also updated the rate constant to 3.6e-10 for this reaction according to IUPAC recommendations. Further, we have fixed C$_2$H$_6$SCl chemistry as proposed, replacing the previous unintended null cycle with the reaction C$_2$H$_6$SCl = DMSO + ClO. These are important fixes, though we do find only small differences in our overall results after implementing these changes.

The OH-addition reaction between OH and DMS gives SO2, MSA and CH3O2 as the products. Quoteing Pham and Spracklen. Looking at Spracklen they have that channel for the DMS oxidation giving 0.6SO2 and 0.4DMSO. The DMSO can then react with OH to give MSA. The mechanism included in the model seems to have lumped this together to avoid having to have DMSO as a tracer. However, there is DMSO as a tracer in place for the oxidation of BrO.

**Response:** This appears to be a typo in the reaction Table 2 of the manuscript. We had this reaction in the model as DMS + OH → 0.60SO$_2$ + 0.4DMSO + CH$_3$O$_2$, and we have corrected Table 2 to address this.

The basis for some of these rates is some rather old complications of recommended rates (Saunders et al., 2003, Burkholder et al., 2015). There are more upto date recommendations in the the literature by both IUPAC and JPL. It would be very useful to update the mechanism to these recommendations rather than relying on some rather elderly rate constants.

**Response:** We acknowledge the need for better rate constants and have updated the kinetics of several reactions according to IUPAC and JPL recommendations along with following more recent literature offering updated reaction kinetics or stoichiometry. For some reactions we have also gone to MCMv3.3.1 values. In total, these revisions have impacted our results by decreasing the global mean surface-layer gas-phase sulfur dioxide (SO$_2$) mixing ratio by 35% compared to 38% and enhancing sulfate aerosol (SO$_4^{2-}$) mixing ratio by 22% compared to 16%, compared to the previously submitted version of the manuscript. Overall, we can say the revised mechanism shows updates in the magnitude of the changes mentioned previously, while maintaining the general direction of changes, along with associated conclusions and narrative. We greatly appreciate the guidance and opportunity to make these improvements. Changes to the manuscript for the reaction table thus to the mechanism are listed here:

| Gas-phase reactions | Rate (s$^{-1}$) | References | Table No. |
|---|---|---|---|
| DMS + OH → 0.60SO$_2$ + 0.4**DMSO** + CH$_3$O$_2$ | $8.2 \times 10^{-39}$[O$_2$]e$^{5376/T}$/(1+1.05× $10^{-5}$([O2]/[M])e$^{3644/T}$) cm$^3$molecule$^{-1}$s$^{-1}$ | (Burkholder et al., 2015; Pham et al., 1995; Spracklen et al., 2005) | 2 |
| DMS + BrO → DMSO + Br | **1.50e-14*exp(1000/T)** | **(Bräuer et al., 2013; Hoffmann et al., 2016)** | 2 |
| DMS + O$_3$ → SO$_2$ | **1.50e-19** | (Burkholder et al., 2015; Du et al., 2007) | 2 |

| | | | |
|---|---|---|---|
| $OOCH_2SCH_2OOH + NO \rightarrow CH_3O_2S + NO_2 + HCHO$ | 4.9e-12*exp(260/T) | (Saunders et al., 2003) | 3 |
| **MSP** + $HO_2 \rightarrow$ **$CH_3SCH_2OOH$** + $O_2$ | **1.13e-13*exp(1300/T)** | **MCMv3.3.1, (Wollesen de Jonge et al., 2021)** | 3 |
| **$CH_3SCH_2OOH$ + hv $\rightarrow$ $CH_3SCH_2O$ +OH** | **J(41)** | **MCMv3.3.1, (Wollesen de Jonge et al., 2021)** | 3 |
| $HPMTF + OH \rightarrow HOOCH_2SCO + H_2O$ | **4.00e-12** | **(Jernigan et al., 2022)** | 3 |
| **HPMTF + OH$\rightarrow$ 0.13OCS + 0.87SO$_2$ + CO** | **1.40e-11** | **(Jernigan et al., 2022)** | 3 |
| **OCS + OH $\rightarrow$ SO$_2$** | **1.13e-13*exp(1200/T)** | **(Jernigan et al., 2022)** | 3 |
| $DMS + Cl \rightarrow 0.45MSP + 0.55C_2H_6SCl + 0.45HCl$ | **3.60e-10** | (Fung et al., 2022; **Enami et al., 2004**) | 4 |
| **$C_2H_6SCl \rightarrow DMSO + ClO$** | **4.00e-18** | **(Hoffmann et al., 2016)** | 4 |
| **MSP + CH$_3$O$_2$ $\rightarrow$** $CH_3SCH_2(O) + O_2$ | 3.74e-12 | (Saunders et al., 2003) | 4 |
|  |  |  | 2 |
|  |  |  | 2 |
|  |  |  | 3 |

| Aqueous-phase reactions | $k_{298}$ [M$^{-1}$s$^{-1}$] | References | Table No. |
|---|---|---|---|
| $DMS\ (aq) + O_3\ (aq) \rightarrow DMSO\ (aq) + O_2\ (aq)$ | $8.61 \times 10^8$ | (Gershenzon et al., 2001; **Hoffmann et al., 2016**) | 2 |
| $DMSO\ (aq) + OH\ (aq) \rightarrow MSIA\ (aq)$ | **$6.65 \times 10^9$** | (Zhu et al., 2003; **Hoffmann et al., 2016**) | 2 |
| $MSIA\ (aq) + OH\ (aq) \rightarrow MSA\ (aq)$ | $6.00 \times 10^9$ | (Sehested and Holcman, 1996; **Hoffmann et al., 2016**) | 2 |
| $MSI^-\ (aq) + OH\ (aq) \rightarrow MSA\ (aq)$ | $1.20 \times 10^{10}$ | (Bardouki et al., 2002; **Hoffmann et al., 2016**) | 2 |
| $MSIA\ (aq) + O_3\ (aq) \rightarrow MSA\ (aq)$ | $3.50 \times 10^7$ | (Hoffmann et al., 2016) | 2 |
| $MSI^-\ (aq) + O_3\ (aq) \rightarrow MSA\ (aq)$ | $2.00 \times 10^6$ | (Flyunt et al., 2001; **Hoffmann et al., 2016**) | 2 |
| $MSA\ (aq) + OH\ (aq) \rightarrow SO_4^{2-}$ | $1.50 \times 10^7$ | **(Hoffmann et al., 2016)** | 2 |
| $MS^-\ (aq) + OH\ (aq) \rightarrow SO_4^{2-}\ (aq)$ | $1.29 \times 10^7$ | (Zhu et al., 2003; **Hoffmann et al., 2016**) | 2 |

Overall, I feel that the new chemistry scheme has rather crudely merged previously developed chemistry schemes without much thought to the underlying assumptions in these scheme. These previous schemes have made various approximations, but the new mechanism doesn't seem to have understood these approximations and developed a scheme which is capable of either removing these approximations or by dealing with them appropriately. It has just patched things on top of each other. It would be advantageous to read the primary literature, the IUPAC and NASA recommendations for rate constants and use these as the basis of creating a consistent mechanism which uses the latest current thinking for this oxidation Unless the mechanism can be better updated and it then better explained I don't think the basis of this work is built on weak foundations.

**Response:** We do appreciate the concern for building a consistent, up-to-date mechanism for DMS oxidation and have taken all these suggestions to heart. Our revised mechanism comprises multiple improvements, including the removal of incorrectly duplicated oxidation reactions, updated kinetics for reactions that have more recent recommendations and, in some cases, better explanations for reaction choices. We hope that these changes make sense and believe that they strengthen the mechanism as recommended. While the overall direction of impacts and final qualitative conclusions remain consistent with previous results, these changes collectively redistribute the relative importance of the sinks for DMS, affect the major products yield such as HPMTF, MSA and $SO_2$, and shift the magnitudes of the global mean surface-layer gas-phase sulfur dioxide ($SO_2$) and sulfate aerosol ($SO_4^{2-}$) as well as aerosol number concentration for all size ranges.

- **DMS and HPMTF Concentrations.**

After developing this new chemistry oxidation scheme the modelled DMS and HPMTF concentrations are compared to those from the ATOM-4 mission. The model does pretty poorly for DMS and surprisingly well for HPMTF. This leaves the authors in a difficult position. The DMS emissions could be wrong, but that would imply that the HPMTF is then right for the wrong reason. The DMS emissions could be right but the DMS lifetime was too long but that would imply an error in the chemistry mechanism. Or the DMS observations could be incorrect. They show the seasonal cycle from one surface site which looks pretty good as additional justification, but this doesn't seem sufficient.

**Response:** We acknowledge that the model seems to compare poorly with DMS observations from the ATom-4 mission, which makes agreement for other species somewhat surprising at face value. Related to this comment, we did find an error in the scripts used for the ATom-4 vertical profile plots of DMS and HPMTF, which has been resolved in the current version. This error did not impact any results or figures other than the vertical profiles (now Figure 3). We have also added boxplots to Figure 3 for various mechanism perturbations to help contextualize the comparison with observations, including one in which HPMTF is not lost to cloud and aerosols, and one for the original BASE GEOS-Chem mechanism.

In Figure 3a of the revised manuscript we see that DMS output is still overpredicted, but much less so than with BASE chemistry. These findings are comparable to other literature sources that have used the same measurements for model/observation comparisons (Fung et al., 2022; Novak et al., 2021). Considering the good agreement with long term surface DMS measurements (Figure 2,

including an additional site), it is also possible that the underlying DMS emissions themselves are reasonable in terms of global seasonal budgets, but poorly resolved on finer spatiotemporal scales, especially at our coarse 4°x5° horizontal grid, leading to some of the differences in agreement seen here.

After resolving the error in our vertical profile plots, the model with full chemistry (MOD) compares poorly with HPMTF observations from the ATom-4 mission, revealing a strong low bias. Without heterogeneous losses of HPMTF to clouds and aerosols (shown in orange in Figure 3) this bias reverses to show a slight overprediction, indicating high sensitivity of the model to these processes. We acknowledge these issues in the text and note the need for additional work constraining these rates and processes.

Finally, we note that the remaining issues with ATom-4 DMS comparisons are consistent with those found in other similar studies, with which our results compare favorably (Novak et al. 2021 and Fung et al. 2022), and that comparisons against DMS and HPMTF are in any case much improved relative to base GEOS-Chem chemistry. While the development of improved DMS emissions and the resolution of HPMTF cloud and aerosol loss rates are outside the scope of this work, we do agree that these improvements should be a high priority for the modeling community and would in turn greatly benefit chemical mechanism evaluation and development.

It would be useful to discuss the DMS emissions in the model more. What is the emission in the model? How does this compare to previous studies? Are the model emissions higher / lower than other studies etc? How much wiggleroom is there here for improving the model performance?

**Response:** To address this comment regarding DMS emissions in GEOS-Chem we have added lines 396-405:

"The global DMS emission flux ($F_{DMS}$) from ocean to the atmosphere is 22 Tg S yr$^{-1}$ and is within the range of 11– 28 Tg S yr$^{-1}$ simulated by GEOS-Chem and other models in previous studies (Lennartz et al., 2015; Spracklen et al., 2005; Hezel et al., 2011, Fung 2022, Chen 2018). Our $F_{DMS}$ is higher than the 18 Tg S yr$^{-1}$ which uses sea surface DMS concentration from Kettle et al. (1999) as reported (Chen et al 2018) indicating the DMS emission varies with change in sea surface DMS climatology. The analysis and improvement of DMS emissions directly is not a part of this work, but we note that improved and validated inventories for DMS will certainly play a role in subsequent oxidation product comparisons. We recommend ongoing evaluation of DMS emissions inputs to complement the expanded chemical mechanism development we present here."

If think that more analysis is needed to show that the DMS concentration calculated by the model are 'reasonable' and that the ATOM DMS observations can be reconciled with the model. I would suggest that more comparisons with surface sites would provide the increased confidence here. It seems difficult to go onto the next stage of the analysis (the impact on aerosols), without having confidence in the ability of the model to get the DMS concentrations right. At the moment there is some doubt.

**Response:** We have added one more surface site (now Figure 2 in the revised version) of long term observations to compare model versus surface observations of DMS, providing further evaluation

of model performance. We find that our DMS oxidation mechanism again does well in comparison with the BASE simulation by narrowing the gap between model and observation for both sites shown in Figure 2. Considering the use of emissions inventories that may poorly represent spatiotemporal distributions of DMS emissions, we believe that our results provide broad support for overall modeled DMS budgets and expanded chemistry, while highlighting biases and uncertainties in the details for exactly where and when that DMS is emitted.

**Minor issues:**

Table 1. Can the rate constants be put into this table?

**Response:** We have added the rate constants in the Table 1.

**Table 1.** The three DMS oxidation reactions in the standard GEOS-Chem chemical mechanism

| Reactions | Rate ($s^{-1}$) | |
|---|---|---|
| $DMS + OH_{(abstraction)} \rightarrow SO_2 + CH_3O_2 + CH_2O$ | $1.20e\text{-}11*exp(-280/T)$ | (R1) |
| $DMS + OH_{(addition)} \rightarrow 0.75\ SO_2 + 0.25\ MSA + CH_3O_2$ | $8.2\times10^{-39}[O_2]e^{5376/T}/(1+1.05\times10^{-5}([O2]/[M])e^{3644/T})$ $cm^3molecule^{-1}s^{-1}$ | (R2) |
| $DMS + NO_3 \rightarrow SO_2 + HNO_3 + CH_3O_2 + CH_2O$ | $1.90e\text{-}13*exp(530/T)$ | (R3) |

Figure 1. Where do the numbers come from for this table. It would be useful to point towards the simulation that is being used?

**Response:** The numbers on the arrows were indicating the production and loss rates in the units of Gg S $yr^{-1}$. The numbers in the box next to the scheme were emission flux of DMS (in Gg S $yr^{-1}$), burdens (GgS) and deposition (Gg S $yr^{-1}$) calculated from different diagnostic outputs of the simulation. However, in the revised version we removed all the numbers from the scheme and put that into a table in the Appendix section, as Table B1 along with flux of DMS, lifetime of major sulfur compounds in the mechanism and global mean concentration of major oxidants. We also rephrased the Figure caption for Figure 1 to include the name of the simulation that is represented by the scheme as:

"**Figure 1** Modified DMS oxidation mechanism used in this work (simulation MOD) showing the formation of major stable oxidation products (blue-outline boxes) including newly identified intermediate HPMTF, and their contribution to new particle formation or growth of existing particles. Note that we include gas-phase and aqueous-phase chemistry of DMSO, MSIA and OCS in the mechanism, which counts towards their contribution to the formation of major oxidation products of DMS, but that these pathways are not explicitly shown here to maintain visual clarity."

Page 6. It might be benefitial to start with a description of the gas phase aspects of the model before moving to the aerosol scheme?

**Response:** We have revised the methodology section by starting with an explanation of the gas-phase aspects of the tropospheric chemistry option in the model, and then introduce the aerosol scheme as part of the size distribution analysis.

Page 8. The literature contains other DMS oxidants (IO, Br, etc) Why were these not included in the scheme? They may be considered small but it would be good to explain that.

**Response:** We have added the reason in lines 448-451:

"Due to slower reaction kinetics and lower fractional contribution reported earlier compared to BrO with DMS and uncertainty in surface concentration and kinetics for photochemically generated halogenated species such as Br, IO we did not include them in our chemical scheme (Chen et al., 2018)."

Page 10. A table of DMS emissions and global (hemispheric sinks) would be useful here. Lifetimes to different oxidants would also provide some useful way of comparing the different oxidation routes in the BASE and the MOD simulations. It would also be useful to provide information on the global (hemispheric) mean concentration of important oxidants (OH, Cl, NO3, O3, BrO etc).

**Response:** We have added all these details in Table B1 of the appendix section.

Line 280. Does the MOD simulation have wet and dry deposition of DMS? Could more information be provided about that?

**Response:** We have added line 271-274 as:

"In all our simulations including MOD, DMS is advected and undergoes chemical loss and transport but does not undergo dry or wet deposition. However, dry and wet deposition of oxidation products such as DMSO, MSIA, MSA and HPMTF are included."

Line 323. I would put the model / measurement comparisons section before the budget details. I would start with an analysis of the model's ability to simulate DMS (both from aircraft and from the group) and then move onto HPMTF.

**Response:** We have reorganized the result discussion section by first introducing the model-observation comparisons for DMS for two different surface sites and then for the ATom-4 aircraft data versus model output of DMS and HPMTF respectively to establish the model's ability to capture these species. Later we move on to the analysis of the overall budget and specific sinks for DMS followed by the implications of our chemical mechanism in influencing the size distribution of resulting aerosol in the final part of the results and discussion section.

Line 388. Is there a 37% reduction in the global $SO_2$ burden with the change of chemistry? Or is that a spatially averaged fractional change?

**Response:** By this we mean a 37% reduction (this was a typo in the submitted manuscript and was supposed to be 38%, now it is 35% in the revised version) in the global surface production of $SO_2$ with the change of chemistry (simulation MOD) compared to the simplified DMS chemistry of BASE simulation (now line 457-458 of the revised manuscript). This is not a spatially average fractional change.

---

## Author Response (AR2)

**Response to Referee 1 (RC1)**

Mechanism
I still have some concerns about the mechanism used. As I said in my previous review it has been pulled together from different mechanisms making different assumptions. It is mainly a puling together of mechanisms in pervious models with the inherent assumptions and implications that they have made.

**Response:** We again appreciate the reviewer's detailed attention to the underlying justifications behind the chemical mechanism we have used for this work. Our newly revised mechanism for this round of review includes multiple improvements, including the addition of new intermediate reactions, updated kinetics for reactions with more recent recommendations, and most importantly original references wherever appropriate. We believe that these changes should satisfy our reviewers' concerns and make the final mechanism a strong replacement for the current simplified DMS oxidation scheme used in the GEOS-Chem model. As with our previous revisions, the overall direction of impacts and final qualitative conclusions remain consistent with our original results. The new set of changes collectively do influence the relative importance of the sinks for DMS, influence the final yield of major products such as HPMTF, MSA and $SO_2$, and shift the magnitudes of the global mean surface-layer gas-phase sulfur dioxide ($SO_2$) and sulfate aerosol ($SO_4^{2-}$) concentrations, as well as aerosol number concentration for all size ranges. Point-by-point responses to other specific comments and concerns are provided below.

It would be useful for the authors to reference their reactions either by the original lab study, from the IUPAC / JPL compilation or when not available indicate that the rate constant has been estimated. For example Novak and Wollesen de Jonge use the DMS+NO3 rate constant but the "original reference" for this comes from the IUPAC recommendation based on rate constant measured in the 1980s. The Saunders et al., 2003 reference doesn't discuss DMS.

**Response:** We acknowledge the value of original references and have thoroughly revised our reference list. We have also updated several reactions following more recent literature that includes updated reaction kinetics or stoichiometry. We have further simplified our notation, replacing the formula $CH_2O$ with HCHO throughout. These revisions have affected the impact of our expanded mechanism compared to the base case. For example, the global mean surface-layer gas-phase sulfur dioxide ($SO_2$) mixing ratio now drops by 40% compared to 35% under the previous version, while the sulfate aerosol ($SO_4^{2-}$) mixing ratio increases on average by 17% compared to 22%. Furthermore, compared to our previous mechanism version, the DMS + OH addition pathway has increased in importance compared with the abstraction pathway, consistent with Cala et. al., 2023. Mechanism and reaction table changes are listed here:

**Table 2.** Overview of the DMS oxidation mechanism via OH-addition pathway.

| Gas-phase reactions | Rate constant $(cm^3\ molecule^{-1}\ s^{-1})$ | References |
|---|---|---|
| DMS + OH →  DMSO + HO₂ | $9.5\times10^{-39}[O_2]exp(5270/T)/(1+ 7.5\times10^{-29}[O_2]exp(5610/T))$ | IUPAC SOx22 (upd. 2006) |
| DMS + BrO → DMSO + Br | $1.50\times10^{-14}exp(1000/T)$ | (Bräuer et al., 2013; Hoffmann et al., 2016) |
| DMS + O₃ → SO₂ | $1.50\times10^{-19}$ | (Du et al., 2007; Burkholder et al., 2020) |
| DMSO + OH → 0.95(MSIA + CH₃O₂) | $6.10\times10^{-12}exp(800/T)$ | MCMv3.3.1, (von Glasow and Crutzen, 2004; Burkholder et al., 2020) |
| MSIA + OH → 0.95(SO₂ + CH₃O₂) | $9.00\times10^{-11}$ | MCMv3.3.1 |
| MSIA + OH → 0.05(MSA + HO₂ + H₂O) | $9.00\times10^{-11}$ | von Glasow and Crutzen, 2004 |
|  |  |  |
| MSIA + NO₃ → CH₃SO₂ + HNO₃ | $1.00\times10^{-13}$ | (von Glasow and Crutzen, 2004; Hoffmann et al., 2016) |

| Aqueous-phase reactions | $k_{298}\ [M^{-1}s^{-1}]$ | References |
|---|---|---|
| DMS (aq) + O₃ (aq) → DMSO (aq) + O₂ (aq) | $8.61\times10^{8}$ | (Gershenzon et al., 2001; Hoffmann et al., 2016) |
| DMSO (aq) + OH (aq) → MSIA (aq) | $6.65\times10^{9}$ | (Zhu et al., 2003; Hoffmann et al., 2016) |
| MSIA (aq) + OH (aq) → MSA (aq) | $6.00\times10^{9}$ | ()(Hoffmann et al., 2016; Herrmann et al., 1998) |
| MSI⁻ (aq) + OH (aq) → MSA (aq) | $1.20\times10^{10}$ | (Bardouki et al., 2002; Hoffmann et al., 2016) |
| MSIA (aq) + O₃ (aq) → MSA (aq) | $3.50\times10^{7}$ | Hoffmann et al., 2016; Herrmann et al., 1998) |
| MSI⁻ (aq) + O₃ (aq) → MSA (aq) | $2.00\times10^{6}$ | (Flyunt et al., 2001; Hoffmann et al., 2016) |
| MSA (aq) + OH (aq) → SO₄²⁻ | $1.50\times10^{7}$ | (Hoffmann et al., 2016; Herrmann et al., 1998) |
| MS⁻ (aq) + OH (aq) → SO₄²⁻ (aq) | $1.29\times10^{7}$ | (Zhu et al., 2003; Hoffmann et al., 2016) |

**Table 3.** Overview of the DMS oxidation mechanism involving HPMTF formation.

| Gas-phase reactions | Rate constant $(cm^3\ molecule^{-1}\ s^{-1})$ | References |
|---|---|---|
| MSP (CH₃SCH₂OO) → OOCH₂SCH₂OOH | $2.2433\times10^{11}exp(-9801.6/T)\times(1.0348\times10^{8}/T^{3})$ | (Berndt et al., 2019; Veres et al., 2020; Wollesen de Jonge et al., 2021) |
| OOCH₂SCH₂OOH → HPMTF (HOOCH₂SCHO) + OH | $6.0970\times10^{11}exp(-9489/T)\times(1.1028\times10^{8}/T^{3})$ | (Berndt et al., 2019; Veres et al., 2020; Wollesen de Jonge et al., 2021) |
| OOCH₂SCH₂OOH + NO → HOOCH₂S + NO₂ + HCHO | $4.9\times10^{-12}exp(260/T)$ | MCMv3.3.1 |
| MSP + HO₂ → CH₃SCH₂OOH + O₂ | $1.13\times10^{-13}exp(1300/T)$ | MCMv3.3.1, (Wollesen de Jonge et al., 2021) |
|  |  |  |
| CH₃SCH₂OOH + OH → CH₃SCHO | $7.03\times10^{-11}$ | MCMv3.3.1 |
| CH₃SCHO + OH → CH₃S + CO | $1.11\times10^{-11}$ | MCMv3.3.1 |

| | | |
|---|---|---|
| HPMTF + OH→ HOOCH$_2$SCO + H$_2$O | $4.00 \times 10^{-12}$ | (Jernigan et al., 2022a) |
| HPMTF + OH→ 0.13OCS + 0.87SO$_2$ + CO | $1.40 \times 10^{-11}$ | (Jernigan et al., 2022a) |
| OCS + OH → SO$_2$ | $1.13 \times 10^{-13}$exp(1200/T) | (Jernigan et al., 2022a) |
| HOOCH$_2$SCO → HOOCH$_2$S + CO | $9.2 \times 10^{9}$exp(-505.4/T) | (Wu et al., 2015) |
| HOOCH$_2$SCO → OH + HCHO + OCS | $1.6 \times 10^{7}$exp(-1468.6/T) | (Wu et al., 2015) |
| HOOCH$_2$S + O$_3$ → HOOCH$_2$SO + O$_2$ | $1.15 \times 10^{-12}$exp(430/T) | (Wu et al., 2015) |
| HOOCH$_2$S + NO$_2$ → HOOCH$_2$SO + NO | $6.0 \times 10^{-11}$exp(240/T) | (Wu et al., 2015) |
| HOOCH$_2$SO + O$_3$ → SO$_2$ + HCHO + OH + O$_2$ | $4.0 \times 10^{-13}$ | (Wu et al., 2015) |
| HOOCH$_2$SO + NO$_2$ → SO$_2$ + CH$_2$O + OH + NO | $1.2 \times 10^{-11}$ | (Wu et al., 2015) |

**Table 4.** Overview of the MSA-producing branch of the H-abstraction pathway of DMS oxidation.

| Gas-phase reactions | Rate constant (cm$^3$ molecule$^{-1}$ s$^{-1}$) | References |
|---|---|---|
| DMS + OH → MSP (CH$_3$SCH$_2$OO) + H$_2$O | $1.12 \times 10^{-11}$exp(-250/T) | IUPAC SOx22 (upd. 2006) |
| DMS + Cl → 0.45MSP + 0.55C$_2$H$_6$SCl + 0.45HCl | $3.60 \times 10^{-10}$ | (Fung et al., 2022; Enami et al., 2004) |
| C$_2$H$_6$SCl → DMSO + ClO | $4.00 \times 10^{-18}$ | (Hoffmann et al., 2016; Urbanski and Wine, 1999) |
| DMS + NO$_3$ → MSP + HNO$_3$ | $1.9 \times 10^{-13}$exp(520/T) | MCMv3.3.1, (Novak et al., 2021; Wollesen de Jonge et al., 2021; Atkinson et al. 2004) |
| MSP + NO → CH$_3$SCH$_2$(O) + NO$_2$ | $4.9 \times 10^{-12}$exp(260/T) | MCMv3.3.1 |
|  |  |  |
| MSP + MSP → 2HCHO + 2CH$_3$S | $1.00 \times 10^{-11}$ | (von Glasow and Crutzen, 2004) |
| CH$_3$SCH$_2$(O) → CH$_3$S + HCHO | $1.0 \times 10^{6}$ | MCMv3.3.1 |
| CH$_3$S + O$_3$ → CH$_3$S(O)  | $1.15 \times 10^{-12}$exp(430/T) | MCMv3.3.1; (Atkinson et al., 2004) |
| CH$_3$S + O$_2$ → CH$_3$S(OO) | $1.20 \times 10^{-16}$exp(1580/T) | MCMv3.3.1; (Atkinson et al., 2004) |
| CH$_3$S + NO$_2$ → CH$_3$SO + NO | $3.00 \times 10^{-12}$exp(210/T) | IUPAC SOx60 (upd. 2006); (Atkinson et al., 2004) |
| CH$_3$SO + O$_3$ → CH$_3$O$_2$ + SO$_2$ | $4.00 \times 10^{-13}$ | IUPAC SOx61 (upd. 2006); (Borissenko et al., 2003) |
| CH$_3$SO + NO$_2$ → 0.75(CH$_3$SO$_2$ + NO) + 0.25(SO$_2$ + CH$_3$O$_2$ + NO) | $1.20 \times 10^{-11}$ | (Borissenko et al., 2003; Atkinson et al., 2004) |
| CH$_3$S(OO) → CH$_3$(O$_2$) + SO$_2$ | $5.60 \times 10^{16}$exp(-10870/T) | (Atkinson et al., 2004) |
| CH$_3$S(OO) → CH$_3$SO$_2$ | $1.00$ | (Campolongo et al., 1999; Hoffmann et al., 2016) |
| CH$_3$S(OO) → CH$_3$S + O$_2$ | $3.50 \times 10^{10}$exp(-3560/T) | MCMv3.3.1 |
| CH$_3$SO$_2$ + O$_3$ → CH$_3$SO$_3$ + O$_2$ | $3.00 \times 10^{-13}$ | MCMv3.3.1; (von Glasow and Crutzen, 2004) |
| CH$_3$SO$_2$ → CH$_3$(O$_2$) + SO$_2$ | $5.00 \times 10^{13}$exp(-9673/T) | MCMv3.3.1; (Barone et al., 1995) |
| CH$_3$SO$_2$ + NO$_2$ → CH$_3$SO$_3$ + NO | $2.20 \times 10^{-11}$ | (Atkinson et al., 2004) |
| CH$_3$SO$_3$ + HO$_2$ → MSA  | $5.00 \times 10^{-11}$ | MCMv3.3.1; (von Glasow and Crutzen, 2004) |
| CH$_3$SO$_3$ → CH$_3$(O$_2$) + H$_2$SO$_4$ | $5.00 \times 10^{13}$exp(-9946/T) | MCMv3.3.1 |
| MSA + OH → CH$_3$SO$_3$ | $2.24 \times 10^{-14}$ | MCMv3.3.1 |

The H abstraction path chemistry I'm a bit confused about. We get to CH3S and then this can primary react with O2 to form CH3S(OO) with a rate constant of 1.2e-16*exp(1580). However, the back reaction for this CH3S(OO)-->CH3S is missing from the reaction scheme but is in the MCM and is substantially faster. This wouldn't matter so much if the only fate of CH3S was reaction with O2 but it can also react with O3. Why is this reaction missing?

**Response:** To address this concern, we have added the reaction $CH_3S(OO) \rightarrow CH_3S + O_2$ with a rate constant of $3.50 \times 10^{10} exp(-3560/T)$, following MCMv3.3.1. In addition, $CH_3S$ in our revised version reacts with $NO_2$ via $CH_3S + NO_2 \rightarrow CH_3SO + NO$ with a rate constant of $3.00 \times 10^{-12} exp(210/T)$, following IUPAC SOx60 (upd. 2006) and Atkinson et al., 2004.

Why isn't the reaction between CH3SH2OOH and OH included as it is the the MCM? Is this not competative against the photolysis? What is J(41)

**Response:** We have added the reaction of $CH_3SH_2OOH$ and OH as follows from Table 3. along with oxidation of the product $CH_3SCHO$ to generate $CH_3S$ based on MCMv3.3.1. In the previous version, the photolysis reaction appears to be a typo in the reaction table, and was not present in the model mechanism itself. Even though there is a photolysis reaction for $CH_3SH_2OOH$ according to MCMv3.3.1, we have decided to not include any photolysis reaction for our current mechanism for simplicity and consistency with comparable works. We do agree that the following two reactions are impactful enough to include, again following the example of similar work (Wollesen de Jonge et al., 2021, Cala et al., 2023):

| Gas-phase reactions | Rate constant (cm³ molecule⁻¹ s⁻¹) | References |
|---|---|---|
| $CH_3SCH_2OOH + OH \rightarrow CH_3SCHO$ | $7.03 \times 10^{-11}$ | MCMv3.3.1 |
| $CH_3SCHO + OH \rightarrow CH_3S + CO$ | $1.11 \times 10^{-11}$ | MCMv3.3.1 |

The mechanism then produces CH3S(OO) which in this mechanism has 2 fates. Decomposition to give CH3(O2) or decomposition to give CH3SO2. So I think CH3(O2) is meant to be CH3O2 as used previously here? The CH3S(OO) in this mechanism can also decompose to give CH3SO2. I don't see this reaction in the MCM (https://mcm.york.ac.uk/MCM/species/CH3SOO). There is a CH3SO2 in the MCM but its formed from CH3SOO2 rather than from CH3SOO. Where does the rate constant of 1s-1 for the rearrangement of CH3S(OO) Into CH3SO2 come from? Can the authors clarify what is going on here?

**Response:** We thank for pointing out the issue with the reference. We did add the original reference for this rearrangement reaction in Table 4 as follows. It is originally from Campolongo et al., 1999 and is further usd by Fung et al., 2022, Hoffmann et al., 2016 and Wollesen de Jonge et al., 2021. We have also added an additional reaction for $CH_3S(OO)$ based on MCMv3.3.1 in the revised version as mentioned below from Table 4:

| Gas-phase reactions | Rate constant (cm$^3$ molecule$^{-1}$ s$^{-1}$) | References |
|---|---|---|
| $CH_3S(OO) \rightarrow CH_3SO_2$ | 1.00 | (Campolongo et al., 1999; Hoffmann et al., 2016) |
| $CH_3S(OO) \rightarrow CH_3S + O_2$ | $3.50 \times 10^{10} \exp(-3560/T)$ | MCMv3.3.1 |

What happens to the CH3SCH2O which is formed from the CH3SCH2OOH photolysis?

**Response:** We do not have the photolysis reaction in the revised manuscript. In our revised mechanism $CH_3SH_2OOH$ oxidizes by OH to $CH_3SCHO$, which further oxidizes by OH to $CH_3S$ according to MCMv3.3.1 as shown in Table 3 along with further oxidation of $CH_3S$ listed in Table 4.

Where the authors have ignored reactions in the MCM chemistry they should indicate why and give new references. It would be useful if the text of the paper described the choices use but the authors in constructing the scheme. What is the basis of the reactions?

**Response:** We have gone back through our mechanism description to add further clarification on original studies for key reactions. Relevant lines are listed below:

Line 194 – 199 in the revised manuscript:
"We further implement and evaluate a custom chemical mechanism for DMS oxidation, referred to as "MOD" (Table 2-4), representing an integration of three individual DMS oxidation mechanism updates explored previously using GEOS-Chem and CAM6-Chem. This mechanism also includes HPMTF loss to clouds and aerosols via heterogenous chemistry, dry and wet deposition of HPMTF, along with further improvement based on recent literature updates to chemical kinetics (Chen et al., 2018; Fung et al., 2022; Veres et al., 2020; Novak et al., 2021; Cala et al., 2023)."

Line 225-247 in the revised manuscript:
"As shown in Table 2, the modified DMS chemistry simulations examined here include gas- and aqueous-phase oxidation of DMS and its intermediate oxidation products by OH, $NO_3$, $O_3$, and halogenated species as previously explored in an older version of GEOS-Chem (Chen et al., 2018). The aqueous-phase reactions in cloud droplets and aerosols were parameterized assuming a first-order loss of the gas-phase sulfur species (Chen et al., 2018). Further building upon this previous mechanism, the scheme used here also includes the formation and loss of HPMTF as previously tested in the global climate model CAM6-Chem as shown in Table 3 (Veres et al., 2020). Table 4 presents the third piece of the mechanism: a gas-phase MSA-producing branch of the H-abstraction pathway in the DMS chemistry bridging the other two sets of the reactions (Fung et al., 2022). To avoid addition of $SO_3$ oxidation chemistry we have replaced $SO_3$ with $H_2SO_4$ followed by previous work for the decomposition reaction of $CH_3SO_3$ (Table 4). A similarly integrated mechanism (Table 2-4) has been previously explored using the CAM6-Chem model with a focus on radiation budget impacts, which is improved in this work through updated rate constants and the inclusion of additional relevant reactions (Fung et al., 2022; Novak et al., 2021; Wollesen de Jonge et al., 2021; Cala et al., 2023). The newly added reactions

and their respective rate constants are largely based on the MCMv3.3.1, along with the literature cited in the Table 2-4 reference list. We use a rate constant of $1.40 \times 10^{-11}$ cm$^3$ molecules$^{-1}$s$^{-1}$ for HPMTF + OH, which was previously determined based on concentrations of other known sulfur species (DMS, DMSO, SO$_2$ and methyl thioformate; MTF; CH$_3$SCHO; a structurally similar proxy to HPMTF) and evaluated by box model (Jernigan et al., 2022a). An exploration of reaction rate uncertainty for the HPMTF+OH reaction (Table 3), including both high and low end limits of $5.5 \times 10^{-11}$ cm$^3$ molecules$^{-1}$s$^{-1}$ and $1.4 \times 10^{-12}$ cm$^3$ molecules$^{-1}$s$^{-1}$ resulted in only minor impacts on the fate of HPMTF and ultimate sulfate formation in our simulations (Novak et al., 2021; Wu et al., 2015)."

Also can there consistent representation of the numbers in the tables. Sometimes the multiplication is represented by a * sometimes x sometime exponentials are represented as exp sometimes as e. There is a missing subscript on the O2 on the addition pathway. Why is there units on this and not the other reactions? The table labels these as rates but they are rate constants. Is there a need to give 5 significant figures on the HPMTF rate constants and 3 on the others?

**Response:** We thank the reviewer for such detailed attention to the representation of the reaction tables and revised the tables based on the suggestions. Multiplications are now represented by '$\times$' symbol. The exponential function is now consistently represented by 'exp', and scientific notation 'e's have been replaced with '$10^x$' wherever needed in Table 1-4. We further addressed the subscript issue for the OH addition pathway of DMS, and have removed the listed unit for this reaction. We have named this column "Rate constant" where many of them have temperature dependent expression. For the HPMTF reaction significant figures, we wanted to maintain the precision of the original reference, and so kept the rate constant values as they originally appeared.

Perhaps the authors could use the model given by the reaction table in https://acp.copernicus.org/preprints/acp-2023-42/acp-2023-42.pdf to give the appropriate sources for the reactions? It would also probably help if they considered this mechanism and thought about whether theirs is consistent?

**Response:** We thank the reviewer for referring us to the UKCA chemistry-climate model. We agree that the referenced UKCA publication is a useful example that we have used as a reference point for our gas-phase mechanism choices.

A diagram of the reaction mechanism would help to clarify the mechanism.

**Response:** We thank the reviewer for this suggestion. To help further clarify our mechanism modifications we have redesigned Figure 1 to include additional details, including improved visual representation of key species and reaction pathways.

Other aspects.
There is quite a large change in DMS burden (38%) from the inclusion of the new mechanism. The BrO only constitutes ~20% of the total loss now. It would be useful to assess how the OH,

NO3, O3, BrO concentrations have changed between the model simulations. How much of the change in the burden is the additional routes and how much is a change in the oxidant concentrations? This isn't clear to me and without some sense of how the oxidants have changed it is hard to tell.

**Response:** To address this concern and question, we have added Figure A7 showing concentrations of the major oxidants OH, $NO_3$, $O_3$ and BrO for MOD and MOD-BASE. To make the comparison of the oxidant concentrations of figure A7 with the values in Table B1, we have updated and changed the units for the oxidants in Table B1 except for OH to make them consistent with each other. We hope that this can resolve the questions regarding changes in oxidant concentration between major simulation BASE and MOD.

[Figure]

**Figure A7** Geographic distribution of mean surface oxidant concentrations for simulation (a) BASE and (b) MOD - BASE. Simulations are described in Table 5.